

# How negative can null energy be in large N CFTs?

Jackson R. Fliss[1]*, Ben Freivogel[2]†, Eleni-Alexandra Kontou[3]‡ and Diego Pardo Santos[3]∘

**1** Department of Mathematics and Theoretical Physics, University of Cambridge,
Wilberforce Rd. CB3 0WA, Cambridge, United Kingdom
**2** GRAPPA and ITFA, University of Amsterdam,
Science Park 904, 1098XH Amsterdam, The Netherlands
**3** Department of Mathematics, King's College London,
Strand, London WC2R 2LS, United Kingdom

* jf768@cam.ac.uk , † b.w.freivogel@uva.nl ,
‡ eleni.kontou@kcl.ac.uk , ∘ diego.pardo@kcl.ac.uk

## Abstract

Smeared null energy has been shown to be bounded from below for free minimally coupled quantum field theories. This is not the case for conformally coupled free bosonic theories where states of unbounded null energy can be constructed by increasing the particle number. Little is known for interacting conformal field theories (CFTs) in dimensions larger than two. In this work we consider states that are superpositions of scalar primary operators or the stress-energy tensor itself in large $N$ CFTs. Within the large $N$ approximation we present arguments that the negative smeared null energy of such states scales at worst as the central charge of the theory, $C_T$. This provides evidence for a general bound for CFTs in $d$-dimensions proportional to the central charge.



# 1 Introduction

Energy conditions play an important role in classical gravity: lower bounds on the pointwise energy density form the basis of geometric theorems constraining classical spacetimes. From the mid 1960's it was known that quantum field theory (QFT) is in general incompatible with non-negativity of pointwise energy density and related quantities [1]. Thus pointwise energy conditions do not generically hold for quantum fields in semiclassical gravity.

An important example of this distinction, and the focus of this work, is the null energy condition (NEC) which states that the stress energy tensor contracted with two null vectors is everywhere non-negative, $T_{--} \geq 0$. While this energy condition holds for most classical fields it is violated in QFT. One generalization of the NEC is the average null energy condition (ANEC), that requires the null energy to be nonnegative when averaged over a complete null geodesic. The ANEC can be used as input for geometric theorems in closed universes [2] and is sufficient to prove topological censorship [3, 4]. There are numerous proofs of ANEC for free quantum scalar fields, for flat spacetimes [5, 6] and curved spacetimes [7]. The proof of Wald and Flanagan [8] takes into account backreaction up to second order in perturbation theory. There are also proofs of ANEC from the generalized second law of black hole ther-

modynamics [9] and proofs from generic interacting QFTs [10, 11]. While ANEC is known to fail for chronal geodesics (that is when there exist two points on the geodesic with one in the future of another) the achronal ANEC [12] has no known counterexamples in the context of self-consistent semiclassical gravity.

Despite the success of ANEC, there are several situations where it cannot be applied. Considering the energy conditions as statements about what kinds of spacetimes are allowed, the natural place to examine their applications is classical relativistic theorems and the construction and sustaining of 'exotic spacetimes'. The ANEC cannot be applied to geodesics initiating from a Cauchy surface, such as in the singularity theorems of Penrose [13] and Hawking [14]. Additionally, as mentioned above, the ANEC integrated over a chronal null geodesic has several counterexamples. Thus ANEC itself does not rule out exotic spacetimes such as long wormholes, for example the one described in Ref. [15]. While the anchronal ANEC has no known counterexamples, there is the further issue that a spacetime may not possess complete globally achronal null geodesics [16]. More generally, ANEC is a restrictive condition in its applications as it doesn't give any sense of the local negative energy allowed in QFT.

As an alternative, one may attempt to establish *quantum energy inequalities* (QEIs) bounding the renormalized expectation value of the stress-energy tensor over a segment of a geodesic or a spacetime volume. QEIs have applications in singularity theorems [17, 18] as well as in restricting long wormholes [19].

While QEIs have been derived for free scalar and fermionic fields to date there is no derived QEI for any interacting QFTs in dimensions larger than two. The focus of this work is establishing bounds of the integrated null energy in a class of interacting *conformal field theories* (CFTs) in greater than two dimensions.

In previous work, we examined the null energy for free scalar fields. The natural extension of the NEC would be the null energy integrated over a null geodesic. However, it was shown [5], that contractions of the stress-tensor over finite null geodesic segments are not bounded from below. So instead, a lower bound of the null energy was derived averaged over two null directions, the double smeared null energy condition (DSNEC) [20]. Schematically

$$\int d^2 x^\pm \, g(x^\pm)^2 \, \langle T_{--} \rangle \geq -\frac{\mathcal{F}}{(\delta^+)^{d/2-1}(\delta^-)^{d/2+1}} \,, \tag{1}$$

where $g(x)^2$ is a smearing function,[1] $\delta^\pm$ are its smearing lengths in the $x^\pm$ directions, $d$ is the spacetime dimension and $\mathcal{F}$ is a dimensionless parameter depending on the details of how the operator is smeared. For massless fields, $\mathcal{F}$ is simply proportional to the number of fields. The lower bound here is derived for free, minimally coupled scalars and it is *state independent*. These kinds of bounds can be used for deriving singularity theorems. Of course, singularity theorems require an integral over a single null geodesic. However, by viewing $\delta^+$ as a small and constant regulator, we can take the DSNEC as a regulated lower bound on along a null geodesic segment. Then one can derive a different bound the smeared null energy condition (SNEC) [20, 21] which can be used to prove the Penrose singularity theorem [18].

When considering fields with non-minimal coupling to gravity though, the situation is different. Those theories include a term in the Lagrangian of the form

$$\delta \mathcal{L} = \xi R \phi^2 \,, \tag{3}$$

---

[1]In this paper a *smearing function* is a smooth square integrable function with square integrable derivatives up to a sufficiently high degree. We will by convention normalize $g(x)$ such that

$$\int d^d x \, g(x)^2 = 1 \,. \tag{2}$$

where $\xi$ is the dimensionless coupling constant. Importantly, these theories include the free bosonic CFT with traceless stress-tensor where $\xi$ is equal to the conformal coupling ($\xi = 1/6$ for four dimensions). Even classically, the NEC is violated in this theory and this violation persists in the limit of vanishing Ricci scalar. Additionally, if one tries to derive a bound for non-minimal coupling to gravity the lower bound is always *state dependent*. For example, in the flat space limit, the DSNEC becomes

$$\int d^2 x^\pm \, g(x^\pm)^2 \, \langle T_{--} \rangle_\psi \geq -\frac{\mathcal{F}}{(\delta^+)^{d/2-1}(\delta^-)^{d/2+1}} - \frac{\# |\xi| \phi_{\max}^2}{(\delta^-)^2}, \tag{4}$$

where $\psi$ is the class of states of bounded field values, and $\phi_{\max}^2$ is the maximum value $\left| \langle \phi^2 \rangle \right|$ can obtain within that class. It also means there are states with unbounded negative null energy simply by increasing the maximum field value by, e.g., increasing the number of particles. Similar results were found for the energy density [22] and the effective energy density [23].

These two results are not independent. When considering free scalar fields, single-particle states can only admit negative energies for some test functions if the classical model violates the energy condition. More formally, consider $\phi_1$ the normalized state vector of this one-particle state and assume we have

$$\langle \phi_1 | \rho \, \phi_1 \rangle < 0, \tag{5}$$

where $\rho$ is the null energy, the energy density or a similar quantity. Then the $n$-particle state vector $\phi_n$ is created by acting with $n$ copies of $\phi_1$. That makes the average energy

$$\langle \phi_n | \rho \, \phi_n \rangle = n \langle \phi_1 | \rho \, \phi_1 \rangle. \tag{6}$$

That means two things: one that it is possible to construct states with unlimited negative energy by increasing the number of particles, and two that there is no state independent bound for the corresponding QEI.

While state-dependent bounds such as (4) are ostensibly weaker than their state-independent counterparts due to the possibility of triviality, (e.g. the bound is the same as the original quantity) they do have utility in the context of effective field theory (EFT). Indeed, semi-classical gravity is at best an EFT. A state-dependent bound then provides a useful diagnosis for the applicability of lower-bounded energy densities in terms of expectation values of simpler operators. For the case above, [24] illustrated how large $|\langle \phi^2 \rangle|$ values signal breakdowns of effective field theory in both the Einstein and Jordan frame.

The above discussions do not readily extend to fermionic fields and interacting fields. The direct study of QEIs in interacting QFTs remains a difficult problem. However, the two dimensional Ising model provides one example where there exists a one-particle state admitting negative energy density but the corresponding QEI has a state-independent lower bound [25]. What lessons we can extrapolate from QEIs of free fields to interacting QFTs, in general, remains an important open question.

In this work, we focus on negative null energy in CFTs, which offers a fruitful area of investigation. CFTs generically lie at the end points of an interacting QFT's renormalization group flow while also providing new structure associated to conformal symmetry. More importantly, the problems discussed above remain relevant for CFTs as well: the free theory of non-minimally coupled fields only admits a state-dependent DSNEC bound even for conformal coupling. Whether this is true for interacting CFTs is not straightforward.

Two dimensional CFTs offer a guidepost for this question. There are results for integrable models [26] and a general result for two dimensional CFTs [27]. The CFTs covered by this result are all models built from unitary highest-weight representations of the Virasoro algebra. Then the smeared null energy is bounded below by

$$\int d\lambda \, g(\lambda)^2 \langle T_{--} \rangle_\psi \geq -\frac{C_L}{12\pi} \int d\lambda \, (g'(\lambda))^2, \tag{7}$$

where the integral is over a null geodesic parametrized by $\lambda$ and $C_L$ is the left-moving central charge of the theory.[2] Here, $g(\lambda)$ is a smooth, real function, which gives a finite bound when it is of compact support (or falls off sufficiently fast). The result was generalized to all spacetimes globally conformal to Minkowski [21].

Does this result hold for $d > 2$? As discussed above, the ANEC holds for interacting CFTs in $d > 2$. The answer for QEIs is still uncertain but we give an indication that a general state-independent bound proportional to the central charge of the theory is possible.

Specifically, the CFTs appearing in this work are *large N* CFTs: they have a large central charge (here defined, across all dimensions, as the coefficient of the stress-energy tensor two-point function),[3] $C_T$ and a class of simple operators, called *single trace*, whose correlation functions factorize, e.g. schematically

$$\langle \mathcal{O}\mathcal{O}\mathcal{O}\mathcal{O} \rangle = \langle \mathcal{O}\mathcal{O} \rangle \langle \mathcal{O}\mathcal{O} \rangle + 1/N^2 \,. \tag{8}$$

Here we will assume there is a single-parameter, $N$, controlling the factorization of all single-trace operators of the theory. This is true of familiar large $N$ theories, however, see [29] for counter-examples. By convention, we relate $N$ to the central charge via

$$C_T = N^2 \,. \tag{9}$$

$N$ also provides a rough count on the number of degrees of freedom; a canonical example to keep in mind is $\mathcal{N} = 4$ super-Yang Mills with $SU(N)$ gauge group. Due to factorization, the single trace primary operators in large $N$ CFTs behave similarly to single-particle operators of free fields. Moreover, they come part-in-parcel with a class of *multi-trace operators* which, in analogy, behave as their corresponding multi-particle operators [30]. Details of this will be provided below.

Additionally, large $N$ CFTs play a special role in the AdS/CFT correspondence [30–32]. Through the holographic dictionary, single trace operators in the CFT map to elementary fields in the bulk and the factorization of single-trace CFT correlation functions is commensurate with the bulk fields being free. The first deviations from full factorization come from tree-level exchanges of bulk (single-trace) fields and the vertices appearing in such exchanges scale universally as $N^{-1} \sim \sqrt{G_N}$.[4] While many of the results of this paper do not rely on our CFTs being holographic *per se*, we will refer to holographic CFTs for intuition at a couple of points.

Thus large $N$ CFTs provide a testing ground for whether the features of free theories that allow unbounded negative null energy densities imply as much as in interacting theories. In this work, we argue that the mechanisms allowing unbounded negative smeared null energy in free CFTs fail in interacting CFTs. To be specific, we investigate smeared negative null energies in two main classes of states: states prepared with scalar primary operators and states prepared with the stress-energy tensor.

- We show that states prepared by acting with scalar primary operators with conformal dimensions $\Delta \geq d$ have positive null energy density. We give evidence that the same result holds for states prepared by a pointwise insertion of the stress-energy tensor in $d \geq 4$, a previously unknown result.

- We show that states prepared with light scalar operators $\Delta < d$, as well as states which are a superposition of the vacuum and a state created by acting with the stress-energy tensor, schematically

$$|\psi\rangle = |\Omega\rangle + T_{\mu\nu}|\Omega\rangle \,, \tag{10}$$

---

[2]Two-dimensional CFTs can have two independent central charges $C_R$ and $C_L$ and thus two different bounds for the two independent null components of the stress-tensor.

[3]This is to distinguish it from conformal anomaly coefficients in even dimensions, e.g. $(a, c)$ in $d = 4$, which are also called the central charges of the theory. The $R^2$ anomaly coefficient, $c$, is proportional to $C_T$ [28].

[4]Though see [33] for recently constructed counter-examples.

can admit negative smeared null energies.

- In large $N$ theories, states prepared by multi-trace operators $\mathcal{O}^p$, with $\mathcal{O}$ a light scalar, as well as those prepared by stress-energy tensors, can have negative null energy densities enhanced by the order $p$ of the multi-trace operator. However, there is a breakdown of factorization of states prepared by sufficiently heavy multi-trace operators. We show, within the framework of large $N$ factorization, that the scaling of the negative smeared null energy is at worst proportional to the central charge of the theory.

- We indicate how to construct states in holographic theories and product CFTs with negative smeared null energy of order $C_T$. We provide arguments for similar scaling for general large $N$ theories.

Our results support the expectations of $d = 4$ CFTs [34] and provide evidence that smeared null energy is better behaved in interacting CFTs than in free theories. This evidence leads us to a bold conjecture on the smeared null energy in (generic) interacting CFTs:

**Strong conjecture:** All interacting CFTs obey a state-independent bound on the smeared null energy of the form

$$\int d^d x \, g(x)^2 \, \langle T_{--}(x)\rangle_\psi \geq -C \, \mathcal{C}_0[g] \,, \tag{11}$$

for all states of the theory. Above, $\mathcal{C}_0[g(x)]$ is a state-independent functional of the smearing function $g(x)$ and $C$ is a constant that scales linearly with the central charges of the theory (see footnote 3). Our current results point towards $C$ being proportional to the stress-energy tensor two-point coefficient, $C_T$, however we allow that this may be a feature of large $N$ theories governed by one parameter of factorization instead of a general feature.

If the strong conjecture were true it would be a powerful statement about CFTs consistently coupled to coupled to gravity, as (11) could provide useful input to singularity theorems beyond the applicability of the ANEC. There is one obvious way the above conjecture could fail. As described above, the free conformally coupled scalar does not satisfy a state-independent bound. This theory does not violate the conjecture because it is free. If it is possible to turn on interactions while preserving conformal invariance, one might expect that the resulting theory will still not satisfy our state-independent bound proposed above. A weaker form of this conjecture which is also allowed by our results is that the true lower bound in interacting CFTs is indeed state-dependent but these state-dependent divergences are also controlled in the class of states we consider. An alternative conjecture suggested by our results is then:

**Weak conjecture:** All CFTs obey a spectrum-dependent lower bound on the smeared null energy where the state dependence is linear and takes the form of the expectation values of a bounded number of operators in the spectrum, i.e.

$$\int d^d x \, g(x)^2 \, \langle T_{--}(x)\rangle_\psi \geq -C \, \mathcal{C}_0[g] + \sum_{\Delta \leq \Delta_\star} \mathcal{C}_\Delta[g] \, \langle \mathcal{O}_\Delta \rangle_\psi \,, \tag{12}$$

for some maximum $\Delta_\star$.

It is natural to suppose that $\Delta_\star = d$ as the higher dimension operators will be less singular than the stress-energy tensor itself. Even if the weak form of the conjecture is true, this is still a useful lower-bound as it would set clear diagnostics on the effective field theories for which energy densities are bounded below in terms of a handful of operator expectation values, *à la* [24].

The paper is organized as follows: In Section 2 we discuss the occurrence of negative null energy in states prepared with scalar operators. We repeat the process in Section 3 for stress-energy tensor states by calculating explicitly the eigenvalues of the stress-tensor three-point

function. In Section 4 we examine the null energy for multi-trace scalar states and continue in Section 5 with examples where the negative null energy is beyond the square root of the central charge. In Section 6 we construct multi stress-tensor states and derive lower bounds of their null energy. We conclude in Section 7 with a summary and discussion of future work.

**Notation**

All the results in this work are on $d$-dimensional Minkowski spacetime unless otherwise specified. As the primary focus of this paper is on the null-energy we will discuss the expectation values of the stress-energy tensor in Lorentzian signature. However we will use the formalism of Euclidean CFT correlators for state preparation; the Lorentzian correlators will then be given by analytic continuation in complex Euclidean time [11, 35]. To this end we delineate notation for Lorentzian and Euclidean signature in what follows. Lorentzian spacetime indices will be indexed by Greek letters, e.g. $x^\mu$ with $\mu = 0, 1, \ldots, d-1$, which can be split into a temporal and spatial components as $x^\mu = (t, \vec{x})$. Spatial positions will be indexed by mid-range Latin letters, e.g. $x^i$, with $i = 1, \ldots, d-1$. Without loss of generality, we will designate null coordinates

$$x^\pm = t \pm x^1, \tag{13}$$

and denote $\vec{x}_\perp^i$ for the remaining coordinates (with $i$ now ranging $i = 2, \ldots, d-1$). The Minkowski metric is given by the 'mostly-plus' signature $\eta_{\mu\nu} = (-1, 1, \ldots, 1)$ so that

$$ds^2 = \eta_{\mu\nu} dx^\mu dx^\nu = -dt^2 + \sum_{i=1}^{d-1} dx^i dx^i = -dx^+ dx^- + \sum_{i=2}^{d-2} dx_\perp^i dx_\perp^i. \tag{14}$$

Euclidean signature is obtained by taking $t \to -i\tau$ and we will denote Euclidean spacetime indices by starting-range Latin letters, e.g. $x^a = (\tau, \vec{x}^i)$. Null coordinates then rotate to $x^\pm \to -i\tau \pm x^1$ which we will denote with complex coordinates[5]

$$(-i\tau + x^1, -i\tau - x^1) \equiv (\bar{z}, -z). \tag{15}$$

The Euclidean distance is then given by

$$ds_E^2 = \delta_{ab} dx^a dx^b = d\tau^2 + \sum_{i=1}^{d-1} dx^i dx^i = dz\, d\bar{z} + \sum_{i=2}^{d-2} dx_\perp^i dx_\perp^i. \tag{16}$$

Because it features so prominently in this work we will make special note of the null stress-energy tensor which in Lorentzian signature is given by $T_{--} = \ell_-^\mu \ell_-^\nu T_{\mu\nu}$ with null-vector $\ell_-^\mu = (1, -1, \vec{0}_\perp)$. Explicitly,

$$T_{--} = T_{00} - 2T_{01} + T_{11}. \tag{17}$$

In Euclidean signature this rotates to

$$T_{zz} = -T_{\tau\tau} - 2i T_{\tau 1} + T_{11}. \tag{18}$$

## 2 Scalar primary states

To discuss the possibility of negative null energy in CFTs we consider unitary CFTs with dimension $d > 2$. We will pay special attention to primary operators, the lowest dimension operators

---

[5]Despite this notational suggestion, it will be important in Euclidean correlators to maintain that $z$ and $\bar{z}$ are independent variables in order to reproduce correct Lorentzian correlators under continuation.

in a given irreducible representation of the conformal algebra. To begin with an illustrative example we first investigate the null-energy density in states prepared by the insertion of a scalar primary operator. We first show how much negative energy is allowed in this state and then find the specific operator that allows for negative null energy.

## 2.1 Neighborhoods of negative null energy

Let $\mathcal{O}_\Delta$ be a primary scalar operator, where $\Delta$ is the conformal dimension, prepared in the lower Euclidean half-plane, $H_- = \{(\tau, \vec{x}) | \tau < 0\}$, by a smooth function of compact support, $h$:

$$|h_\Delta\rangle = \mathcal{N} \int_{H_-} d^d x\, h_\Delta(x) \mathcal{O}_\Delta(x) |\Omega\rangle, \tag{19}$$

where $\mathcal{N}$ normalizes the state and $\Omega$ is the conformal vacuum.

Our observable of interest is the null energy defined as the stress-energy tensor contracted on one null direction. If one of the null directions is $x^-$ and the other $x^+$, we will compute the expectation value of $T_{--}$ in the state (19). To estimate whether the null-energy smeared over a connected neighborhood is positive or negative (and to what degree) it will be sufficient to look at its pointwise expectation value at the origin:

$$\langle T_{--}(0)\rangle_{h_\Delta} = \frac{\int_{H_-} d^d x_1\, d^d x_2\, h_\Delta(x_1)^* h_\Delta(x_2) \langle \mathcal{O}_\Delta(x_1)^\dagger\, T_{--}(0) \mathcal{O}_\Delta(x_2)\rangle}{\int_{H_-} d^d x_1\, d^d x_2\, h_\Delta(x_1)^* h_\Delta(x_2) \langle \mathcal{O}_\Delta(x_1)^\dagger\, \mathcal{O}_\Delta(x_2)\rangle}. \tag{20}$$

We have been careful with the placement of the dagger in taking Hermitian conjugation. To be more specific, we regard

$$\mathcal{O}_\Delta(x)^\dagger = \left( e^{-i\hat{P}_a x^a} \mathcal{O}_\Delta(0) e^{i\hat{P}_a x^a} \right)^\dagger, \tag{21}$$

where $P_a$ is the Euclidean generator of translations. This is contrasted with

$$\mathcal{O}_\Delta^\dagger(x) = e^{-i\hat{P}_a x^a} \mathcal{O}_\Delta^\dagger(0)\, e^{i\hat{P}_a x^a}. \tag{22}$$

This difference would be immaterial in Lorentzian signature (due to reality of $x^\mu$ and Hermiticity of $\hat{P}_\mu$). However, because $\mathcal{O}_\Delta$ has been prepared in Euclidean signature, Hermitian conjugation then acts by reflecting the operator position about a codimension-one Euclidean plane. We will take this plane to be at Euclidean time $\tau = 0$ and so, e.g.

$$\mathcal{O}_\Delta(\tau, \vec{x})^\dagger = \mathcal{O}_\Delta^\dagger(-\tau, \vec{x}), \tag{23}$$

where the $\dagger$ on the inside acts on the internal details of the operator (e.g. if it is a complex operator or possesses Euclidean spacetime indices). If we denote $(\tau, \vec{x}) \equiv x$ we denote its Euclidean reflection as $(-\tau, \vec{x}) \equiv x^R$.

Following an argument by Ref. [22, 24], suppose there exists a scalar operator $\mathcal{O}_\Delta$ and preparation function $h_\Delta(x)$ such that the null energy density at the origin, (20), is negative

$$\langle T_{--}(0)\rangle_{h_\Delta} = -\rho_0, \tag{24}$$

for a fixed $\rho_0 \geq 0$. Then by continuity of the expectation value, for any $\alpha > 0$ there exists a spacetime neighborhood $\mathcal{U}$ containing the origin such that

$$-\alpha \rho_0 \leq \langle T_{--}(x)\rangle_{h_\Delta} \leq -\alpha^{-1} \rho_0, \qquad \forall\, x \in \mathcal{U}. \tag{25}$$

This implies that $T_{--}(x)$ integrated over a positive, normalized smearing function, $g(x)^2$, with compact support in $\mathcal{U}$ is strictly negative and bounded by

$$-\alpha\rho_0 \leq \int d^d x \, g(x)^2 \, \langle T_{--}(x)\rangle_{h_\Delta} \leq -\alpha^{-1}\rho_0 \,. \tag{26}$$

Due to the simple action of scaling on conformal primary operators

$$\lambda^D \, \mathcal{O}_\Delta(x)\lambda^{-D} = \lambda^\Delta \, \mathcal{O}_\Delta(\lambda x) \tag{27}$$

(where $D$ is the dilatation generator) and the scale invariance of the vacuum, it is easy to show that

$$\langle T_{--}(\lambda x)\rangle_{h_\Delta(x)} = \lambda^{-d}\langle \lambda^D T_{--}(x)\lambda^{-D}\rangle_{h_\Delta(x)} = \lambda^{-d}\langle T_{--}(x)\rangle_{\lambda^{d-\Delta}h_\Delta(\lambda x)} \,, \tag{28}$$

by commuting $\lambda^{\pm D}$ through the $\mathcal{O}_\Delta$'s and changing integration variables. Thus by choosing a new preparation function $h'_\Delta(x) \equiv \lambda^{d-\Delta}h_\Delta(\lambda x)$ we can push the smeared null-energy arbitrarily low, however only for smearing functions with support in smaller spacetime neighborhoods:

$$-\lambda^d\alpha\rho_0 \leq \int d^d x \, g(x)^2 \, \langle T_{--}(x)\rangle_{h'_\Delta} \leq -\lambda^d\alpha^{-1}\rho_0 \,, \qquad \text{supp } g(x)^2 \subset \lambda^{-1}\mathcal{U}\,. \tag{29}$$

This is fitting with our normal expectations: although the integrated null-energy density can be negative its magnitude is inversely proportional to the size of the spacetime region it is integrated against.

## 2.2 Types of scalar operators

Let us now discuss what types of scalar operators can allow negative smeared null energy. Taking $\mathcal{O}_\Delta$ to be a real scalar, evaluating the expectation value (20) is straight-forward: the two-point and three-point functions of primary operators are completely fixed by conformal representation theory [28]:

$$\langle \mathcal{O}_\Delta(x_1)^\dagger T_{--}(0)\mathcal{O}_\Delta(x_2)\rangle = \frac{C_{T\mathcal{O}\mathcal{O}}}{|x_1|^{2\Delta_0}|x_2|^{2\Delta_0}\left|x_1^R - x_2\right|^{2(\Delta-\Delta_0)}}\left(\frac{x_1^1 + i\tau_1}{x_1^2} - \frac{x_2^1 - i\tau_2}{x_2^2}\right)^2 \,, \tag{30}$$

and

$$\langle \mathcal{O}_\Delta(x_1)^\dagger \mathcal{O}_\Delta(x_2)\rangle = \frac{1}{\left|x_1^R - x_2\right|^{2\Delta}} = \frac{1}{((\tau_1 + \tau_2)^2 + (\vec{x}_1 - \vec{x}_2)^2)^\Delta} \,. \tag{31}$$

Above, $\Delta_0 = \frac{d-2}{2}$ is the lowest scalar conformal dimension allowed by unitarity and $C_{T\mathcal{O}\mathcal{O}}$ is a three-point coefficient. In our conventions it is negative definite:[6]

$$C_{T\mathcal{O}\mathcal{O}} = -\frac{d}{d-1}\frac{\Delta}{\Omega_{d-1}} \,, \tag{32}$$

where $\Omega_{d-1} = \frac{2\pi^2}{\Gamma(d/2)}$ is the area of the unit $(d-1)$-sphere [28]. The denominator of (20)

$$\int_{H_-} d^d x_1 \, d^d x_2 \, h_\Delta(x_1)^* h_\Delta(x_2)\langle \mathcal{O}_\Delta(x_1)^\dagger \mathcal{O}_\Delta(x_2)\rangle = \int_{H_-} d^d x_1 d^d x_2 \frac{h_\Delta(x_1)^* h_\Delta(x_2)}{\left|x_1^R - x_2\right|^{2\Delta}} \,, \tag{33}$$

is both sign-definite and positive by virtue of being the norm of a state when $\Delta$ is greater than the unitarity bound $\Delta_0$. However for later purposes it will be useful for us to prove positivity

---

[6]The sign of $C_{T\mathcal{O}\mathcal{O}}$ is correlated with the sign of the averaged null energy condition (ANEC) in primary states [36], which has proven to be non-negative in relativistic QFTs [10,11].

directly from the integral (33) which we do in Appendix A. We now focus on the numerator of (20) and make the change of coordinates

$$y^a = \frac{x^a}{x^2},\tag{34}$$

in which the numerator takes the form

$$\int_{H_-} d^d x_1 d^d x_2\, h_\Delta(x_1)^* h_\Delta(x_2) \langle \mathcal{O}_\Delta(x_1)^\dagger T_{--}(0) \mathcal{O}_\Delta(x_2)\rangle = C_{T\mathcal{O}\mathcal{O}} \int_{H_-} d^d y_1\, d^d y_2\, \hat{h}_\Delta(y_1)^* \hat{h}_\Delta(y_2)$$
$$\times \frac{(\bar{z}_{y_1} - z_{y_2})^2}{\left| y_1^R - y_2 \right|^{2(\Delta - \Delta_0)}},\tag{35}$$

where $\hat{h}_\Delta(y^a) = \frac{h_\Delta(y^a/y^2)}{|y|^{2(d-\Delta)}}$. Assuming that $h_\Delta$ does not have support at $\tau = 0$ (which would present problems for the norm of state) we can write this as

$$\int_{H_-} d^d x_1 d^d x_2\, h_\Delta(x_1)^* h_\Delta(x_2) \langle \mathcal{O}_\Delta(x_1)^\dagger T_{--}(0) \mathcal{O}_\Delta(x_2)\rangle = -\frac{C_{T\mathcal{O}\mathcal{O}}}{(\Delta - \Delta_0 - 1)(\Delta - \Delta_0 - 2)}\tag{36}$$
$$\times \int_{H_-} d^d y_1\, d^d y_2\, \frac{\left( \partial_{\bar{z}_{y_1}} \hat{h}_\Delta(y_1) \right)^* \partial_{\bar{z}_{y_2}} \hat{h}_\Delta(y_2)}{\left| y_1^R - y_2 \right|^{2(\Delta - \Delta_0 - 2)}}.$$

This integral is now of the form obtained from the norm of a state prepared by a hypothetical scalar primary operator of dimension $\Delta - \Delta_0 - 2$ and smeared with the function $\partial_{\bar{z}} \hat{h}_\Delta$. This integral is then positive when this hypothetical operator is heavier than the unitarity bound set by $\Delta_0$, i.e. when

$$\Delta \geq d.\tag{37}$$

While this is an intuitive criteria for the sign-definiteness of (36), we directly prove positivity of integrals of this form in Appendix A without the assumption of them arising from operator norms.

What this exercise indicates is that states prepared by sufficiently heavy (i.e. marginal or irrelevant) scalar operators have positive null energy densities that cannot be changed by picking a creative smearing function $h(x)$. We can contrast this with relevant operators for which it may be possible to engineer states of negative null energy.

A salient example of this is given in the free scalar theory with a null stress-energy tensor given by

$$T_{--}(x) =: \partial_-\phi(x)\partial_-\phi(x) : -\frac{d-2}{4(d-1)}\partial_-^2 : \phi(x)^2 :,\tag{38}$$

where normal-ordering is with respect to the creation-annihilation operators of the free theory. The scalar primary saturates the unitarity bound, $\Delta_\phi = \Delta_0 < d$, and so can possibly evade our argument for positivity. This is indeed the case. In [24] it was shown that single-particle states of the form

$$\int d^d x\, h_{\Delta_0}(x)\phi(x)|\Omega\rangle,\tag{39}$$

with preparation function

$$h_{\Delta_0}(x) = \int_0^\infty \frac{dk_-}{2\pi} \int \frac{d^{d-2}k_\perp}{(2\pi)^{d-2}} e^{ik\cdot x} \left[ \sqrt{k_-} \frac{\left(\frac{3}{2}\alpha_- - k_-\right)}{\alpha_-^{3/2}\alpha_\perp^{\Delta_0}} e^{-k_-/\alpha_- - |k_\perp|/\alpha_\perp} \right],\tag{40}$$

can generate negative null-energy densities for all values of real parameters $(\alpha_-, \alpha_\perp)$.

Finally we point out that because the operator product with the stress-energy tensor does not mix conformal modules of scalar primaries, three point functions of the form $\langle \mathcal{O}_{\Delta_1}(x_1) T_{--}(0) \mathcal{O}_{\Delta_2}(x_2) \rangle$ vanish unless $\Delta_1 = \Delta_2$ for scalar operators [37]. Thus we cannot engineer large negative null-energies from superpositions of the form

$$|\psi\rangle = \gamma_1 |h_{\Delta_1}\rangle + \gamma_2 |h_{\Delta_2}\rangle, \tag{41}$$

which simply have additive null-energy:

$$\langle T_{--}(0)\rangle_\psi = \frac{|\gamma_1|^2 \langle T_{--}(0)\rangle_{h_{\Delta_1}} + |\gamma_2|^2 \langle T_{--}(0)\rangle_{h_{\Delta_2}}}{|\gamma_1|^2 + |\gamma_2|^2}. \tag{42}$$

The upshot of this section is that the nature of negative null energy densities in a CFT is spectrum dependent. States prepared by irrelevant and marginal scalar operators ($\Delta \geq d$) have positive null energy densities. For relevant scalar operators, positivity is not guaranteed and it may be possible to generate negative null energy by a suitable preparation function, $h_\Delta$. Regardless, this negative null energy is controlled: due to the scaling properties of primary operators largely negative smeared null energy can only be realized on sufficiently small spacetime neighborhoods.

# 3 Stress-energy tensor states

The spectrum of a CFT is not universal, however there is one operator guaranteed to exist in every local CFT: the stress-energy tensor itself. Thus we will investigate its utility in generating negative null energy states.

## 3.1 Preparation of states

Here we consider states prepared by the Euclidean integration of

$$|h^{ab}\rangle = \int_{H_-} d^d x \, h^{ab}(x) T_{ab}(x) |\Omega\rangle, \tag{43}$$

where $h^{ab}(x)$ is a polarization tensor with smooth compactly supported components. Furthermore, since the two-point function $\langle T_{ab} T_{--}\rangle$ is generically non-vanishing, it is also possible for us consider superpositions with the CFT vacuum of the form

$$|\psi\rangle = \mathcal{N}\left( |\Omega\rangle + \int_{H_-} d^d x \, h^{ab}(x) T_{ab}(x) |\Omega\rangle \right), \tag{44}$$

where $\mathcal{N}$ is the normalization factor. Such superpositions will play an important role in what follows. Now, we consider the null energy by smearing with a smooth compactly supported test function $g$ in the state $\psi$

$$\int d^d x \, g(x)^2 \langle T_{--}(x)\rangle_\psi. \tag{45}$$

To evaluate the negativity or positivity of $\int g^2 \langle T_{--}\rangle_\psi$, it suffices to analyze the pointwise expectation value at the origin

$$\begin{aligned}
\langle T_{--}(0)\rangle_\psi = |\mathcal{N}|^2 \Bigg( &\int_{H_-} d^d x \, h^{ab}(x)^* \langle T_{ab}(x)^\dagger T_{--}(0)\rangle + \int_{H_-} d^d x \, h^{cd}(x) \langle T_{--}(0) T_{cd}(x)\rangle \\
&+ \int_{H_-} d^d x \, d^d y \, h^{ab}(x)^* h^{cd}(y) \langle T_{ab}(x)^\dagger T_{--}(0) T_{cd}(y)\rangle \Bigg),
\end{aligned} \tag{46}$$

with $\langle T_{--}(t)\rangle_\Omega = 0$. The normalization factor is given by

$$\frac{1}{|\mathcal{N}|^2} = 1 + \int_{H_-} d^d x d^d y \, h^{ab}(x)^* h^{cd}(y) \langle T_{ab}(x)^\dagger T_{cd}(y)\rangle. \tag{47}$$

The two-point and three-point functions in (46) can be evaluated using the conformal symmetry of the theory and the conservation of the stress-energy tensor; see Appendix B. The two-point function is fully determined up a positive constant $C_T$, the central charge of the theory

$$\langle T_{ab}(x)T_{cd}(0)\rangle = \frac{C_T}{x^{2d}}\mathcal{I}_{ab,cd}(x), \tag{48}$$

where $\mathcal{I}_{ab,cd}(x)$ is the inversion tensor; see (B.1). We are making an assumption of no contact terms, however for all practical purposes in what follows, stress-energy tensor two and three point functions will be at separated points and do not change our analyses. For $d \geq 3$, the three-point function of the stress-energy tensor evaluated at any three spacetime points is fully fixed up to three independent parameters (two independent parameters for $d = 3$) which depend on the CFT under consideration [28]:

$$\langle T_{ab}(x_1)T_{cd}(x_2)T_{ef}(x_3)\rangle = \frac{\mathcal{I}_{ab,a'b'}(x_{13})\mathcal{I}_{cd,c'd'}(x_{23})t_{a'b'c'd'ef}(X)}{|x_{12}|^d|x_{13}|^d|x_{23}|^d}, \tag{49}$$

where $x_{ij} = x_i - x_j$, $X = \frac{x_{13}}{(x_{13})^2} - \frac{x_{23}}{(x_{23})^2}$ and $t_{abcdef}(X)$ is an homogeneous tensor of degree zero in $X$, symmetric and traceless on each pair of indices $ab$, $cd$ and $ef$; see (B.9). Using this expression, the pointwise null energy (46) is

$$\begin{aligned}
\frac{\langle T_{--}(0)\rangle_\psi}{|\mathcal{N}|^2} =& C_T \int_{H_-} d^d x \frac{h^{ab}(x)^*}{|x|^{2d}}\mathcal{I}^\dagger_{ab,--}(x^R) + C_T \int_{H_-} d^d x \frac{h^{ab}(x)}{|x|^{2d}}\mathcal{I}_{ab,--}(x) \\
&+ \int_{H_-} d^d x_1 d^d x_2 \frac{h^{ab}(x_1)^* h^{cd}(x_2)}{|x_1^R|^d|x_1^R-x_2|^d|x_2|^d}\mathcal{I}^\dagger_{ab,pq}(x_1^R-x_2)\mathcal{I}_{--,lm}(-x_2) \\
&\times t_{pqlmcd}\left(\frac{x_1^R-x_2}{(x_1^R-x_2)^2}+\frac{x_2}{x_2^2}\right),
\end{aligned} \tag{50}$$

where $\mathcal{I}_{ab,--}(x) = -\mathcal{I}_{ab,\tau\tau}(x) + \mathcal{I}_{ab,11}(x) - 2i\mathcal{I}_{ab,\tau 1}(x)$ and $\mathcal{I}^\dagger_{ab,\tau\tau}(x^R) = \mathcal{I}_{ab,\tau\tau}(x)$, $\mathcal{I}^\dagger_{ab,\tau 1}(x^R) = -\mathcal{I}_{cd,\tau 1}(x^R)$ and $\mathcal{I}^\dagger_{ab,ij}(x^R) = \mathcal{I}_{ab,ij}(x)$. The first line integrals in (50) can be negative due to the flexibility in choosing $h^{ab}$. The sign of the second integral is significantly challenging to determine analytically due to the complex combination of the components of $t_{pqlmcd}$.

To address the sign of the second integral, in the next section, we focus on the analysis of this integral for states prepared by $T_{ab}$ localized at a point. By doing a conformal transformation, we can evaluate this contribution in a collinear frame, where the calculation considerably simplifies. Under these conditions, we find that the $\langle TTT\rangle$ contribution to the null energy is non-negative.

## 3.2 Eigenvalues of three-point function in the collinear frame

Consider a state created by acting with some components of the stress-energy tensor at a point in Euclidean spacetime, a localized version of the states in (43)

$$|\Phi\rangle = \mathcal{N} h^{ab} T_{ab}(\tau, \vec{x})|\Omega\rangle, \tag{51}$$

where $\mathcal{N}$ is a normalization. We would like to know whether the expectation value of the null energy is positive semidefinite in these states. We can use translation invariance to place the null energy operator at the origin, so we want to evaluate

$$\langle\Phi|\,T_{--}\,|\Phi\rangle = (h^{ab})^{*}h^{cd}\,\langle\Omega|\,T_{ab}^{\dagger}(-\tau,\vec{x})T_{--}(0)T_{cd}(\tau,\vec{x})\,|\Omega\rangle\,. \tag{52}$$

The three-point function simplifies considerably when the three points are collinear. We would like to show that the general three-point function above can be mapped, using the global conformal group, to a collinear three-point function.

The key point we will demonstrate briefly is that the conformal mapping can be chosen so that it does not scramble the indices of the null stress-energy tensor, $T_{--}(0)$. Therefore the general expectation value (52) is equal to the expectation value in collinear frame,

$$(h^{ab})^{*}h^{cd}\left\langle T_{ab}^{\dagger}(-\tau,\vec{x})T_{--}(0)T_{cd}(\tau,\vec{x})\right\rangle = (\tilde{h}^{ab})^{*}\tilde{h}^{cd}\left\langle T_{ab}^{\dagger}(-\tilde{\tau},0)T_{--}(0)T_{cd}(\tilde{\tau},\vec{0})\right\rangle\,, \tag{53}$$

for suitably redefined $\tilde{h}^{ab}$. The transformation is given by the special conformal transformation,

$$\tilde{x}^{a} = \frac{x^{a}-b^{a}x^{2}}{1-2b_{a}x^{a}+b^{2}x^{2}}\,. \tag{54}$$

Our points of interest are $(\pm\tau,\vec{x})$. Choose

$$b^{\tau}=0\,,\qquad \vec{b}=\frac{\vec{x}}{\tau^{2}+\vec{x}^{2}}\,. \tag{55}$$

This transformation maps

$$(\pm\tau,\vec{x})\rightarrow\left(\pm\frac{\tau^{2}+\vec{x}^{2}}{\tau},\vec{0}\right)\,, \tag{56}$$

and leaves the origin invariant.

Finally, we want to check how the indices transform in $T_{--}(0)$. Expanding the special conformal transformation near the origin gives

$$x'^{a}=x^{a}+\mathcal{O}(x^{2})\,. \tag{57}$$

Using the tensor transformation law shows that the $\mathcal{O}(x^{2})$ terms do not contribute, so our transformation leaves $T_{--}(0)$ invariant.

Therefore, within the class of states created by acting with a local insertion of the stress-energy tensor in Euclidean spacetime, without loss of generality we can focus on states defined along a line passing by the origin. In this collinear frame, the expression for the three-point function is considerably simplified [28]

$$\langle T_{ab}(x_{1})T_{cd}(x_{2})T_{ef}(x_{3})\rangle = \frac{1}{|\tau_{1}-\tau_{2}|^{d}|\tau_{1}-\tau_{3}|^{d}|\tau_{2}-\tau_{3}|^{d}}\mathcal{A}_{abcdef}\,, \tag{58}$$

where $x_{1,2,3}^{a}=(\tau_{1,2,3},\vec{0})$, $\mathcal{A}_{abcdef}=\mathcal{A}_{cdabef}=\mathcal{A}_{efabcd}$, and $\mathcal{A}_{abcdef}$ is symmetric and traceless on each pair of indices, see (B.13). As in the general frame, the tensor $\mathcal{A}_{abcdef}$ is defined up to three parameters that characterize the CFT we are working with.

Therefore, the null energy (53) is

$$\frac{\langle T_{--}(0)\rangle_{\Phi}}{|\mathcal{N}|^{2}} = \frac{h^{ab*}h^{cd}}{2^{d}|\tau|^{3d}}\mathcal{B}_{ab--cd}\,, \tag{59}$$

where $\tau$ is a negative Euclidean time and the matrix $\mathcal{B}_{abcdef}$ is related to $\mathcal{A}_{abcdef}$ (58) by

$$\begin{aligned}\mathcal{B}_{\tau\tau--cd}&=\mathcal{A}_{\tau\tau--cd}\,,\\\mathcal{B}_{\tau i--cd}&=-\mathcal{A}_{\tau i--cd}\,,\\\mathcal{B}_{ij--cd}&=\mathcal{A}_{ij--cd}\,.\end{aligned} \tag{60}$$

The sign of the null energy is governed by the eigenvalues of the matrix $\mathcal{B}_{ab--cd}$. The matrix below shows $\mathcal{B}_{ab--cd}$ where the indices $ab$ and $cd$ correspond to the rows and columns, respectively,

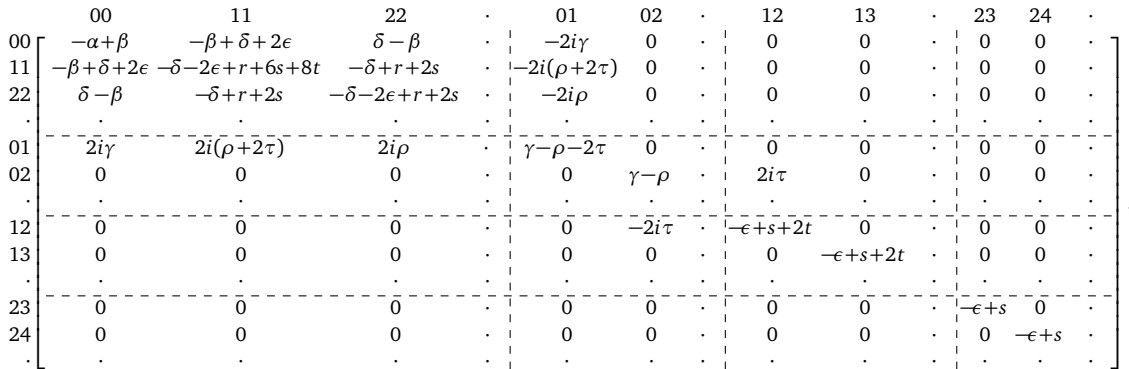

The coefficients in this matrix are fully fixed up to three independent parameters; see Appendix B.2.2.

The normalization factor for this class of localized states (51) is

$$\frac{1}{|\mathcal{N}|^2} = h^{ab*}h^{cd}\langle T_{ab}(\tau)^\dagger T_{cd}(\tau)\rangle = \frac{C_T}{(2\tau)^{2d}}h^{ab*}h^{cd}\mathcal{I}_{ab,cd}^\dagger(\tau). \tag{61}$$

Since our spacetime points in this frame have only time component, $\mathcal{I}_{ab,cd}^\dagger$ is a constant matrix with degenerate eigenvalues 0 and 1 and $|\mathcal{N}|^2 \geq 0$ since $C_T > 0$.

## 3.3 Constraints on the eigenvalues of the three-point functions

For general dimensions $d$, we have CFTs defined by free scalars and free fermions. In even dimensions, $d = 2n$, Abelian $(n-1)$-form potentials yield another free conformally invariant theory. For $d$ odd, there are potentially additional parity-odd structures allowed in the stress-tensor three-point function; for the purposes of this paper we will assume the theories in question preserve parity. In Appendix B.3.1, we summarize the expression for the parameters that characterize these theories [28, 38]. Now we can use the free theories of bosons, fermions, and tensor fields as a basis to express the $\langle TTT \rangle$ of a general interacting theory in even dimensions. We have [34, 39]

$$\langle TTT \rangle = n_s\langle TTT \rangle_s + n_f\langle TTT \rangle_f + n_t\langle TTT \rangle_t, \tag{62}$$

where $\langle TTT \rangle_s$, $\langle TTT \rangle_f$ and $\langle TTT \rangle_t$ are the three-point functions for the theories of free scalar, fermion and tensor fields, respectively, and $n_s$, $n_f$ and $n_t$ are now arbitrary real coefficients. In addition, Hofman-Maldacena bounds [40] derived from the positivity of the averaged null energy condition (ANEC) in any unitary CFT [11,35,39], lead to the following constraints (see Appendix B)

$$-\frac{(d^2-4)d^3}{d-3}n_t \leq 0,$$

$$\frac{1}{2}(d+2)d^2 n_f \geq 0, \tag{63}$$

$$-\frac{(d^2-4)d^2}{2(d-1)^3}n_s \leq 0.$$

For $d \geq 3$, constraints (63) imply that $(n_s, n_f, n_t)$ are positive or zero. The scalar and fermion structures exist for all dimensions, however for $d = 3$, $n_t = 0$. For general odd dimensions there is a third independent parameter. We will still write the three independent parameters

Table 1: Eigenvalues of $\mathcal{B}_{ab--cd}$ for bosonic and fermionic free theories in $d = 3$ normalized by $\Omega_{d-1}/n_{s,f}$.

| Bosons | 21.6348 | 8.0581 | 3.9685 | 0.0982 | -0.0095 | 0 | 0 | 0 | 0 |
|---|---|---|---|---|---|---|---|---|---|
| Fermions | 60.5821 | 24.5062 | 6.2395 | 4.1316 | 0 | 0 | 0 | 0 | 0 |

as $(n_b, n_s, n_t)$ via the relations in Appendix B though $n_t$ might not correspond to a free field structure. From here we focus the analysis of (59) for the free bosonic structure across different dimensions $d$.

### $d = 3$ dimensions

The eigenvalues of the matrix $\mathcal{B}_{ab--cd}$ for $d = 3$ bosonic and fermionic free theories are shown in Table 1. The bosonic theory has a negative eigenvalue, suggesting that it is possible to construct states based on the stress-energy tensor where the three-point function contribution to the pointwise null energy is negative. This case is particularly noteworthy, as it is the only theory we have found with a negative eigenvalue in the collinear frame, as we will show later. Similarly, Ref. [41] also found negative values for the three-point function of the stress-energy tensor in this theory. We should note however that even in this case the ANEC is satisfied, which we numerically verified.

### $d \geq 4$ dimensions

Figure 1 presents the eigenvalues of $\mathcal{B}_{ab--cd}$ for a free bosonic field in dimensions $4 \leq d \leq 20$. Zeros and degenerate eigenvalues are not included. Unlike the $d = 3$ case, all of the eigenvalues are non-negative. We have found the same non-negativity for the eigenvalues of free fermionic and tensor theories. Although this is not a proof, it indicates that we can expect non-negativity of the $\langle TTT \rangle$ contribution to (59) for dimensions $d \geq 4$ for localized states.

The upshot of the above analyses is that in $d \geq 4$ states defined by pointwise insertions of the stress-energy tensor seem to have positive null-energy densities. Thus superpositions of the form (44) are needed to created negative smeared null energy.

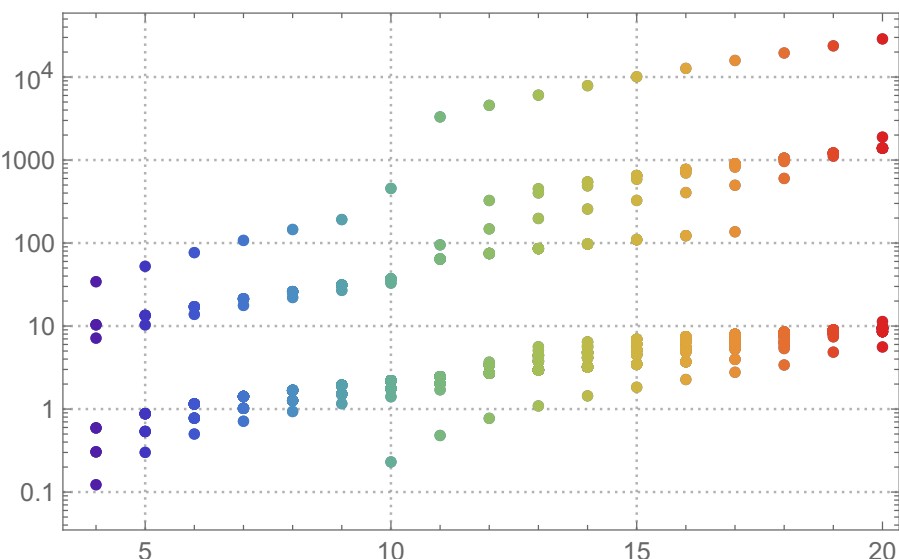

Figure 1: Eigenvalues of the $\mathcal{B}_{ab--cd}$ as a function of the dimension in logarithmic scale.

# 4 Negative null energy in multi-trace states at large $N$

We now indicate how to utilize the properties of large $N$ CFTs to generate large amounts of smeared null energy. The key property that we will rely on is the large $N$ factorization of correlation functions. To warm up we return to our example of primary scalar operators. As we will soon explain, this factorization implies the existence of a particular class of *multi-trace primary operators* whose properties mimic and generalize the notion of normal ordered multi-particle composite operators of free QFTs. More specifically we will review how the consistency of large $N$ factorization with the operator product expansion (OPE) of a primary operator, $\mathcal{O}$, with itself implies the existence of primary operators with OPE coefficients that lead at large $N$ and whose conformal dimension $\Delta$ deviates from the sum $2\Delta_{\mathcal{O}}$ by an anomalous dimension $\gamma$ that vanishes at large $N$.

## 4.1 Large $N$ approximations

Large $N$ factorization is the statement that there is a class of primary operators, called *single-trace operators*, whose correlation functions factorize to leading order in $N$. See [30] for useful review. By convention we can always normalize the coefficient of two-point functions. In this normalization

$$\langle \mathcal{O}_1 \mathcal{O}_2 \rangle \sim 1, \qquad \langle \mathcal{O}_1 \mathcal{O}_2 \mathcal{O}_3 \rangle \sim N^{-1}, \qquad \langle \mathcal{O}_1 \mathcal{O}_2 \mathcal{O}_3 \mathcal{O}_4 \rangle_{\text{conn}} \sim N^{-2}, \tag{64}$$

where

$$\langle \mathcal{O}_1 \mathcal{O}_2 \mathcal{O}_3 \mathcal{O}_4 \rangle_{\text{conn}} := \langle \mathcal{O}_1 \mathcal{O}_2 \mathcal{O}_3 \mathcal{O}_4 \rangle - \langle \mathcal{O}_1 \mathcal{O}_2 \rangle \langle \mathcal{O}_3 \mathcal{O}_4 \rangle - \langle \mathcal{O}_1 \mathcal{O}_3 \rangle \langle \mathcal{O}_2 \mathcal{O}_4 \rangle - \langle \mathcal{O}_1 \mathcal{O}_4 \rangle \langle \mathcal{O}_2 \mathcal{O}_3 \rangle. \tag{65}$$

Above we have dropped all functional dependence of these correlators, focusing instead on their overall $N$-scaling. We have made the assumption that the three-point functions of single-trace operators are all suppressed universally as $N^{-1}$. This is true of familiar examples of large $N$ CFTs however not necessarily so [29, 33]. This assumption ensures that all single-trace operators appearing in an OPE exchange contribute equally to the connected four-point function as the stress-energy tensor itself.

Equation (64) indicates two important facts (i) that at large $N$ the leading contribution to the single-trace four-point functions comes from the disconnected contribution of two-point functions and (ii) the three-point function is disproportionately suppressed (having no $O(N^0)$ contribution). These two facts, (i) the dominance of disconnected lower-point functions and (ii) the relative suppression of odd-point functions, continue to apply to higher-point functions.

The structure of the OPE can be written as [37, 42]

$$\mathcal{O}^A_{\Delta_1}(x) \mathcal{O}^B_{\Delta_2}(0) = \langle \mathcal{O}^A_{\Delta_1}(x) \mathcal{O}^B_{\Delta_2}(0) \rangle \mathbb{1} + \sum_{\Delta_3} \lambda_{\Delta_1 \Delta_2 \Delta_3} \mathcal{P}^{AB\,\Delta_3}_{C\ \ \Delta_1, \Delta_2}(x; \partial) \mathcal{O}^C_{\Delta_3}(0), \tag{66}$$

where the sum is over all non-identity primary operators (labeled by $\Delta_3$) and $(A, B, C)$ schematically index possible $SO(d)$ representations of the operators. The coefficients $\lambda$ are related to three-point function coefficients and are data specifying the CFT. The function $\mathcal{P}$ sums the contributions of descendant operators within the same module as $\mathcal{O}^C_{\Delta_3}$ and is completely fixed by conformal invariance, e.g. through comparing the universal forms of the two-point and three-point functions primary operators:

$$\langle \mathcal{O}^A_{\Delta_1}(x) \mathcal{O}^B_{\Delta_2}(0) \mathcal{O}^C_{\Delta_3}(y) \rangle = \lambda_{\Delta_1 \Delta_2 \Delta_3} \mathcal{P}^{AB\,\Delta_3}_{D\ \ \Delta_1, \Delta_2}(x; \partial^{(y)}) \langle \mathcal{O}^D_{\Delta_3}(0) \mathcal{O}^C_{\Delta_3}(y) \rangle. \tag{67}$$

From this point we will focus on the OPE chanels of identical scalar primaries. To see the necessary existence of the multi-trace primaries consider the four-point function in the $1 \rightarrow 2$

and $3 \rightarrow 4$ OPE channel. Schematically this takes the form

$$\langle \mathcal{O}_\Delta(x_1)\mathcal{O}_\Delta(x_2)\mathcal{O}_\Delta(x_3)\mathcal{O}_\Delta(x_4)\rangle = \langle \mathcal{O}_\Delta(x_1)\mathcal{O}_\Delta(x_2)\rangle \langle \mathcal{O}_\Delta(x_3)\mathcal{O}_\Delta(x_4)\rangle$$
$$+ \sum \lambda^2_{\Delta\Delta\Delta'} \mathcal{P}_A{}^{\Delta'}_{\Delta\Delta} \mathcal{P}_B{}^{\Delta'}_{\Delta\Delta} \langle \mathcal{O}^A_{\Delta'}, \mathcal{O}^B_{\Delta'}\rangle. \tag{68}$$

Comparing to the leading disconnected contribution to the four-point function in (64) and (65), we see that the cross-contractions, $\langle \mathcal{O}_1\mathcal{O}_3\rangle\langle\mathcal{O}_2\mathcal{O}_4\rangle$ and $\langle\mathcal{O}_1\mathcal{O}_4\rangle\langle\mathcal{O}_2\mathcal{O}_3\rangle$ are accounted for by the sum over non-identity primaries. The OPE coefficient is related to the three-point function, and so from (64) we see that the single-trace operators in this sum are suppressed to $O(N^{-2})$. The leading terms, with coefficients $\lambda \sim O(N^0)$, are then a new class of *double-trace operators*, $[\mathcal{O}^2_\Delta]_{k,l}$ with conformal dimension and spin $s$

$$\Delta^{(2)}_{k,l} = 2\Delta + 2k + l + \gamma^{(2)}_{k,l}, \qquad s = l. \tag{69}$$

where $k, l \in \mathbb{Z}_{\geq 0}$ and $\gamma_{k,l}$ is called the *anomalous dimension*. When $k = l = 0$ we will denote this simply by $[\mathcal{O}^2_\Delta]$ and drop the subscripts from the conformal/anomalous dimensions as well. Double-trace operators be regarded as a generalization of normal-ordering of large $N$ single-trace operators,

$$[\mathcal{O}^2_\Delta]_{k,l} \sim \ :\mathcal{O}_\Delta(\partial^2)^k\partial^{\mu_1}\dots\partial^{\mu_l}\mathcal{O}_\Delta: , \tag{70}$$

because they arise from subtracting off self-contractions. We can easily determine their OPE coefficients as well as argue for the vanishing of the anomalous dimensions at large $N$ by looking at three-point functions. For instance, focussing on the lowest-twist (the twist being $\Delta - s$), lowest-spin double trace operator $[\mathcal{O}^2_\Delta]$:

$$\langle \mathcal{O}_\Delta(x_1)\mathcal{O}_\Delta(x_2)[\mathcal{O}^2_\Delta](x_3)\rangle = \lambda_{\Delta\Delta\Delta^{(2)}}\mathcal{P}^{\Delta^{(2)}}_{\Delta\Delta}(x_1 - x_2; \partial_2)\langle[\mathcal{O}^2_\Delta](x_2)[\mathcal{O}^2_\Delta](x_3)\rangle$$
$$= 2\langle \mathcal{O}_\Delta(x_1)\mathcal{O}_\Delta(x_3)\rangle\langle\mathcal{O}_\Delta(x_2)\mathcal{O}_\Delta(x_3)\rangle + O(N^{-2}). \tag{71}$$

Where in the first line we performed the $1 \rightarrow 2$ OPE and in the second we evaluated the contribution of $[\mathcal{O}^2]$ to the four-point function to leading order in $N$. Comparing the leading terms[7] as $x_1 \rightarrow x_2$:

$$\lambda_{\Delta\Delta\Delta^{(2)}}\frac{1}{|x_2 - x_3|^{4\Delta}}\left(1 + \gamma^{(2)}\log\frac{|x_1 - x_2|}{|x_2 - x_3|^2} + \dots\right) = \frac{2}{|x_2 - x_3|^{2\Delta}} + O(N^{-2}), \tag{73}$$

we see that $\lambda_{\Delta\Delta\Delta^{(2)}} = 2$ and $\gamma^{(2)} = O(N^{-2})$. The same exercise can be repeated for the remaining double-trace operators at higher twist and spin.

We can similarly argue for the existence of an entire class of multi-trace primaries

$$[\mathcal{O}^p_\Delta] \sim :\underbrace{\mathcal{O}_\Delta\,\mathcal{O}_\Delta\dots\mathcal{O}_\Delta}_{p}:, \tag{74}$$

formed by the successive OPE with the single-trace $\mathcal{O}_\Delta$. For the simplicity of our argument we will focus on the lowest twist and lowest spin multi-trace operators from this point. Assuming the existence of a multi-trace primary $[\mathcal{O}^p_\Delta]$ with $\Delta^{(p)} = p\Delta + \gamma^{(p)}$ with $\gamma^{(p)} \sim O(N^{-2})$ and whose correlation functions with other primary operators are given by factorization, e.g.

$$\langle \mathcal{O}_\Delta(x_1)\dots\mathcal{O}_\Delta(x_p)[\mathcal{O}^p_\Delta](y)\rangle = p!\,\langle\mathcal{O}_\Delta(x_1)\mathcal{O}_\Delta(y)\rangle\dots\langle\mathcal{O}_\Delta(x_p)\mathcal{O}_\Delta(y)\rangle + O(1/N), \tag{75}$$

---

[7]The expansion of $\mathcal{P}^{\Delta_3}_{\Delta_1\Delta_2}(x,\partial)$ begins with

$$\mathcal{P}^{\Delta_3}_{\Delta_1\Delta_2}(x,\partial) = \frac{1}{|x|^{\Delta_1+\Delta_2-\Delta_3}}(1+\dots). \tag{72}$$

then in the OPE of $\mathcal{O}_\Delta$ and $[\mathcal{O}_\Delta^p]$ there exists a multi-trace primary operator of lowest twist and spin, $[\mathcal{O}_\Delta^{p+1}]$, with dimensions $\Delta^{(p+1)} = (p+1)\Delta + \gamma^{(p+1)}$ with $\gamma^{(p+1)} \sim O(N^{-2})$ and obeying the same factorization properties:

$$\mathcal{O}_\Delta(x_1)[\mathcal{O}_\Delta^p](x_2) \supset \lambda_{\Delta,\Delta^{(p)},\Delta^{(p+1)}} \mathcal{P}_{\Delta\Delta^{(p)}}^{\Delta^{(p+1)}}(x_1 - x_2; \partial_2)[\mathcal{O}_\Delta^{(p+1)}](x_2). \tag{76}$$

By evaluating the three-point function $\langle [\mathcal{O}_\Delta]^{(p)}(x_1)\mathcal{O}_\Delta(x_2)[\mathcal{O}_\Delta^{(p+1)}](x_3)\rangle$ both through the OPE and through large $N$ factorization in the $1 \to 2$ limit, we learn that

$$\lambda_{\Delta\Delta^{(p)}\Delta^{(p+1)}} = p+1, \qquad \Delta^{(p+1)} = (p+1)\Delta + \gamma^{(p+1)}, \qquad \gamma^{(p+1)} = O(N^{-2}), \tag{77}$$

similar to before.

## 4.2 States with large negative null energy density

As indicated in Section 2.2, when a scalar operator, $\mathcal{O}_\Delta$, is sufficiently light ($\Delta < d$) it is possible to prepare a state through smearing $\mathcal{O}_\Delta$ in Euclidean time with an appropriately chosen smearing function:

$$|\mathcal{O}_h\rangle = \int d^d x \, h_\Delta(x)\mathcal{O}_\Delta|\Omega\rangle, \tag{78}$$

such that the smeared null energy density is negative on some neighborhood, $\mathcal{U}$:

$$\int d^d x \, g(x)^2 \langle T_{--}(x)\rangle_{\mathcal{O}_h} = -\bar{\rho}, \tag{79}$$

for some $\bar{\rho} \geq 0$ and supp $g(x)^2 \subset \mathcal{U}$. Let us now assume that $\mathcal{O}_\Delta$ is single-trace in the large $N$ sense. Then for any $p$ we define the state

$$|\mathcal{O}_h^p\rangle = \mathcal{N}(\mathcal{O}_h)^p|\Omega\rangle, \tag{80}$$

prepared by the following integrated multi-trace operator:

$$(\mathcal{O}_h)^p := \int \left(\prod_{i=1}^p d^d x_i h_\Delta(x_i)\right)\left(\prod_{i=1}^{p-1} \lambda_{\Delta\Delta^{(i)}\Delta^{(i+1)}} \mathcal{P}_{\Delta\Delta^{(i)}}^{\Delta^{(i+1)}}(x_{i+1}-x_i; \partial_i)\right)[\mathcal{O}_\Delta^p](x_1), \tag{81}$$

and where $\mathcal{N}$ is a normalization. The operator, (81), is arrived at by taking the subsequent OPEs of $h_\Delta\mathcal{O}_\Delta$ and projecting onto the lowest-twist scalar multi-trace module of the OPE, (76), at each iteration.

Relying on the fact that $T_{ab}$ is, itself, a single-trace operator, (a fact that we will go into more detail on in the following section), the evaluation of the null stress-tensor in this state is easily facilitated by large $N$ factorization:

$$\begin{aligned}\langle T_{--}(x)\rangle_{(\mathcal{O}_h)^p} &= \frac{p^2(p-1)! \langle \mathcal{O}_h^\dagger T_{--}(x)\mathcal{O}_h\rangle \left(\langle \mathcal{O}_h^\dagger \mathcal{O}_h\rangle\right)^{p-1}}{p!\left(\langle \mathcal{O}_h^\dagger \mathcal{O}_h\rangle\right)^p} + O(1/N) \\ &= p\langle T_{--}(x)\rangle_{\mathcal{O}_h} + O(1/N).\end{aligned} \tag{82}$$

Thus for any $g(x)^2$ with support of the original neighborhood $\mathcal{U}$, this state has a negative smeared null energy of

$$\int d^d x \, g(x)^2 \langle T_{--}(x)\rangle_{(\mathcal{O}_h)^p} = -p\,\bar{\rho} + O(1/N). \tag{83}$$

## 4.3   How negative can it be?

The result (83) indicates that given an integrated single-trace state with small amount of negative null energy density, it is possible to amplify this null energy density by piling on single-trace operators in the OPE. This is exactly the situation as in free-field theory [24]: given a one-particle state of negative null energy density, the corresponding multi-particle state proportionally amplifies that null energy density. In the free theory this amplification is unbounded: barring other constraints imposed on effective field theory, the particle number can be made arbitrarily large and the null energy density is unbounded below even if smeared.

In our large $N$ theory, however, we cannot simply pile single-trace operators *ad-infinitum* and expect to obtain the same results. This is because for $p$ large enough, various assumptions leading to the calculation of (83) break down. There are multiple possible avenues for this breakdown, the most pertinent being mixing of multi-trace operators with a large number of single-trace primaries or large corrections to their anomalous dimensions. In this paper we will take the simplest diagnostic and ask when is the leading factorization into two-point functions challenged by the combinatorics of pulling out connected higher-point functions. For instance consider the norm of the state $|\mathcal{O}_h^p\rangle$. Aside from factorizing into two-point functions, the next-to-leading order in factorization could pull out a single connected-four point function:

$$
\begin{aligned}
\langle (\mathcal{O}_h)^{p\dagger}(\mathcal{O}_h)^p \rangle = {}& p! \langle \mathcal{O}_h^\dagger \mathcal{O}_h \rangle^p + \binom{p}{2}^2 (p-2)! \langle (\mathcal{O}_h)^{2\dagger}(\mathcal{O}_h)^2 \rangle_{\text{conn}} \langle \mathcal{O}_h^\dagger \mathcal{O}_h \rangle^{p-2} + \dots \\
& \sim p! + p(p-1)p! N^{-2} + \dots ,
\end{aligned}
\tag{84}
$$

where in the second line we have only indicated the $N$ scaling.

We could also ask about the combinatorics of pulling out two three-point functions, e.g.

$$
\begin{aligned}
\langle (\mathcal{O}_h)^{p\dagger}(\mathcal{O}_h)^p \rangle \supset {}& \binom{p}{2} p \binom{p-1}{2} (p-2)(p-3)! \langle \mathcal{O}_h^\dagger (\mathcal{O}_h^2) \rangle \langle (\mathcal{O}_h)^{2\dagger} \mathcal{O}_h \rangle \langle \mathcal{O}_h^\dagger \mathcal{O}_h \rangle^{p-3} \\
& \sim p(p-1)(p-2)p! N^{-2} ,
\end{aligned}
\tag{85}
$$

which has the same scaling in $N$ as the connected four-point function. However, because $(\mathcal{O}_h)^2$ is *defined* as an (integrated) primary operator, it has no overlap with $\mathcal{O}_h$. Because the leading factorization only defines $[\mathcal{O}_\Delta^2]$ through the $\mathcal{O}_\Delta \mathcal{O}_\Delta$ OPE to leading order in $N$, this requirement fixes an order $N^{-1}$ mixing with the single-trace operator $\mathcal{O}_\Delta$. From here on we will assume that we have fixed all connected odd-point functions to vanish in the computation of the $(\mathcal{O}_h)^p$ two-point functions.

Dominance of the two-point functions in (84) then indicates that the maximum scaling of $p$ that allows the computation leading to (83) is

$$
p_{\max} \sim N ,
\tag{86}
$$

and leading to a maximum scaling of the negative null energy density of

$$
\int d^d x \, g(x)^2 \langle T_{--}(x) \rangle_{(\mathcal{O}_h)^{p_{\max}}} = -N \, \bar{\rho} = -C_T^{1/2} \bar{\rho} ,
\tag{87}
$$

where we have alternatively stated this in terms of the central charge of the CFT.

# 5   Beyond $\sqrt{C_T}$ negative null energy

We do not expect that a bound scaling as $\sqrt{C_T}$ is fundamental; instead it represents a limitation on the regime of validity of large $N$ factorization. In this section let us demonstrate some possible mechanisms by which the smeared null energy can be more negative. In all cases though it doesn't scale more than linearly with the central charge.

## 5.1 Contributions from connected correlators

Above, we have restricted the order of multi-trace operators building our states by looking for the first deviation in large $N$ factorization. This first deviation appears when a connected four-point function becomes combinatorically favorable to two two-point functions.

By permitting ourselves to calculate connected correlators, we will be able to relax the restriction on the order of the multi-trace operators preparing our states and potentially increase the magnitude of the negative null-energy. We can illustrate this first by looking at the leading terms in the inclusion of connected four-point function to

$$\langle T_{--}(x)\rangle_{(\mathcal{O}_h)^p} = \frac{p^2 \langle \mathcal{O}_h^\dagger T_{--}(x)\mathcal{O}_h\rangle \langle (\mathcal{O}_h)^{p-1\dagger}(\mathcal{O}_h)^{p-1}\rangle}{\langle (\mathcal{O}_h)^{p\dagger}(\mathcal{O}_h)^p\rangle} + \dots \tag{88}$$

The sum of all connected four-point and two-point correlators is

$$\langle (\mathcal{O}_h)^{p\dagger}(\mathcal{O}_h)^{p\dagger}\rangle\Big|_{2,4-\mathrm{pt}} = p!\langle \mathcal{O}_h^\dagger \mathcal{O}_h\rangle^p \sum_{m=0}^{\lfloor\frac{p}{2}\rfloor} \frac{p!}{m!(p-2m)!}\mathsf{x}^m, \qquad \mathsf{x} = \frac{\langle (\mathcal{O}_h)^{2\dagger}(\mathcal{O}_h)^2\rangle_{\mathrm{conn}}}{4\langle \mathcal{O}_h^\dagger \mathcal{O}_h\rangle^2}, \tag{89}$$

and overall the expectation value is given by

$$\langle T_{--}(x)\rangle_{(\mathcal{O}_h)^p} = p\langle T_{--}(x)\rangle_{\mathcal{O}_h} \times \mathsf{R}(\mathsf{x},p), \qquad \mathsf{R}(\mathsf{x},p) = \frac{\sum_{m=0}^{\lfloor\frac{p-1}{2}\rfloor} \frac{(p-1)!}{m!(p-2m-1)!}\mathsf{x}^m}{\sum_{m=0}^{\lfloor\frac{p}{2}\rfloor} \frac{p!}{m!(p-2m)!}\mathsf{x}^m}, \tag{90}$$

see Appendix C for details. Since $\mathsf{x} \sim O(N^{-2})$, every term in the numerator and denominator sums appearing in $\mathsf{R}$ would be suppressed if $p$ scales with $N$ with any power less than one. However suppose that $p \sim N^\alpha$ with $\frac{4}{3} > \alpha > 1$ (we will shortly derive this upper bound on $\alpha$). Then the combinatoric terms can quickly dominate the suppression from the connected correlator itself. For example, for the $m \sim O(1)$ terms in the sum

$$\frac{p!}{m!(p-2m)!}\mathsf{x}^m \sim p^{2m}\mathsf{x}^m \sim N^{2m(\alpha-1)}, \tag{91}$$

which grows quickly with increasing $m$. Finding the maximal term in either the numerator or the denominator is difficult, furthermore we cannot strictly bound $\mathsf{R}$ without knowing if $\mathsf{x}$ is positive or negative. Instead we argue in Appendix C that $\mathsf{R} \to 1$ as $p$ grows large with $p \sim N^\alpha$ for the current range of $\alpha$. This is indicated in Fig. 2. This indicates to us that given a smearing function, $g(x)^2$ and preparation function $h_\Delta$ such that (79) is true, then even with the inclusion of connected four-point functions we have

$$\int d^d x\, g(x)^2 \langle T_{--}(x)\rangle_{(\mathcal{O}_h)^{p_{\max}}} \sim -N^{4/3}\bar{\rho} \sim -C_T^{2/3}\bar{\rho}. \tag{92}$$

We have taken $p_{\max} \sim N^\alpha$ at the upper bound of $\alpha$. This upper bound is set by favoring two-point and connected four-point correlators over connected six-point correlators. The number of ways to pull out a single connected six-point correlator is

$$\langle (\mathcal{O}_h)^{p\dagger}(\mathcal{O}_h)^p\rangle \supset \left(p!\langle \mathcal{O}_h^\dagger \mathcal{O}_h\rangle^p\right)\frac{p!}{(p-3)!}\mathsf{x}_3, \qquad \mathsf{x}_3 := \frac{\langle (\mathcal{O}_h)^{3\dagger}(\mathcal{O}_h)^3\rangle_{\mathrm{conn}}}{(3!)^2\langle \mathcal{O}_h^\dagger \mathcal{O}_h\rangle^3}. \tag{93}$$

Since $\mathsf{x}_3 \sim N^{-4}$ this competes with the leading two-point function factorization when

$$p_{\max} \sim N^{4/3}. \tag{94}$$

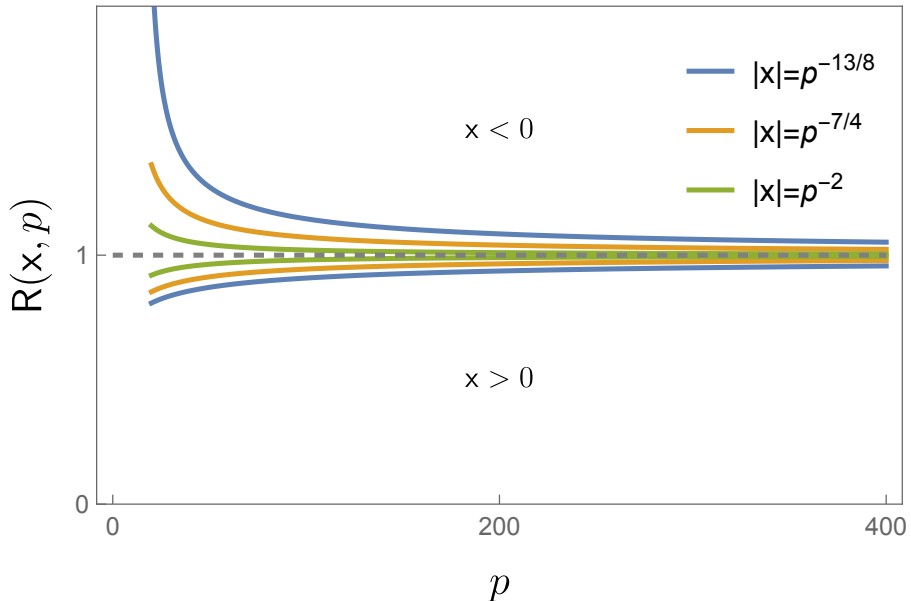

Figure 2: Numerical plots of $R(x, p)$ for large $p$. Since $x \sim N^{-2}$ each plot corresponds to $p \sim N^{\alpha}$ with $1 \leq \alpha < \frac{4}{3}$ (specifically $\alpha = 1, \frac{8}{7}, \frac{16}{13}$, from bottom to top in the legend, respectively). When $x > 0$ the $R$ approaches 1 from below while when $x < 0$, it approaches 1 from above.

We can speculate on how far this can be pushed. If we are willing to keep connected $(2a)$-point correlators up to some $a_{max}$, then a leading contribution to our multi-trace state takes the form

$$\langle T_{--}(x) \rangle_{(\mathcal{O}_h)^p} = p \langle T_{--}(x) \rangle_{(\mathcal{O}_h)} \times R_{a_{max}}(\{x_a\}, p) + \dots , \tag{95}$$

where

$$R_{a_{max}}(\{x_a\}, p) = \frac{\sum_{\{m_a\}}^{p-1} \frac{(p-1)!}{(p - \sum_{a=2}^{a_{max}} a m_a - 1)!} \left[ \prod_{a=2}^{a_{max}} \frac{x_a^{m_a}}{m_a!} \right]}{\sum_{\{m_a\}}^{p} \frac{p!}{(p - \sum_{a=2}^{a_{max}} a m_a)!} \left[ \prod_{a=2}^{a_{max}} \frac{x_a^{m_a}}{m_a!} \right]} , \tag{96}$$

where $\sum_{\{m_a\}}^{p}$ is shorthand for summing over a collection of $m_a$ natural numbers such that $\sum_{a=2}^{a_{max}} a m_a \leq p$, and

$$x_a = \frac{\langle (\mathcal{O}_h)^{a\dagger} (\mathcal{O}_h)^a \rangle_{conn}}{(a!)^2 \langle \mathcal{O}_h^\dagger \mathcal{O}_h \rangle^a} \sim N^{2-2a} . \tag{97}$$

See Appendix C for details. As argued in that appendix, $R_{a_{max}}$ approaches 1 for large $p \sim N^{\alpha}$. Similar to above, the range of validity for $\alpha$ can be determined by allowing connected $(2a_{max})$-point functions but requiring connected $(2a_{max}+2)$-point functions to remain subleading. This happens for

$$p_{max} \sim N^{\frac{2a_{max}}{a_{max}+1}} . \tag{98}$$

So for at least the term of focus in (95) we get

$$\int d^d x\, g(x)^2 \langle T_{--}(x) \rangle_{(\mathcal{O}_h)^p} \sim -N^{\frac{2a_{max}}{a_{max}+1}} \bar{\rho} + \dots \sim -C_T^{\frac{a_{max}}{a_{max}+1}} \bar{\rho} + \dots , \tag{99}$$

which gets closer to $-C_T$ as $a_{max}$ gets larger. Of course this is only schematic since we have ignored terms where $T_{--}$ itself appears in a higher-point connected correlator. But this is strong indication that by allowing more access to the microscopic CFT data (in this case, in the form of connected high-point correlators) one can build states with close to order $C_T$ negative smeared null energy.

## 5.2 Product CFTs

Consider a theory obtained from the tensor product of $M$ identical 'seed' CFT's:

$$(\text{CFT})_{\text{total}} = (\text{CFT})_{\text{seed}}^{\otimes M} . \tag{100}$$

The stress-energy tensor of the product theory is the sum of the stress-energy tensors. Given a state of the seed theory with negative null energy, we can put each of the copies in an identical state, $|\psi\rangle_{\text{total}} = |\psi\rangle_{\text{seed}}^{\otimes M}$. This will yield a total null energy density of

$$\left\langle T_{--}^{\text{total}} \right\rangle = M \left\langle T_{--}^{\text{seed}} \right\rangle . \tag{101}$$

Additionally, $C_T$ of the product theory scales linearly with $M$ as well,

$$C_T^{\text{tot}} = M C_T^{\text{seed}}, \tag{102}$$

since the stress-energy tensors of the seed theories do not interact with each other.

First, this gives a construction of a family of states where the null energy scales linearly with the central charge,

$$\int d^d x \, g(x)^2 \left\langle T_{--}^{\text{total}}(x) \right\rangle \sim -\# C_T^{\text{tot}} . \tag{103}$$

We can also ask whether these product states saturate the lower bound, i.e. could there be non-product states that scale with a larger power of $M$ and hence of $C_T$? This is impossible. The stress-energy tensor is a sum of the seed stress-energy tensors, so the eigenstates of the smeared total stress-energy tensors are product states. Therefore, the lower bound of the smeared total stress-energy tensor *is* a product state.

Taken together, we conclude that for product CFT's,

$$\int d^d x \, g(x)^2 \left\langle T_{--}^{\text{total}}(x) \right\rangle \geq -C_T^{\text{tot}} \mathcal{C}[g], \tag{104}$$

where $\mathcal{C}[g]$ is a $C_T$-independent constant that depends on the smearing function $g$. Free theories can be roughly thought of as a special case of product CFT's.

## 5.3 Holographic theories

Another special case is holographic theories. As mentioned in the introduction, Section 1, in the context of AdS/CFT, single-trace CFT operators are related through the holographic dictionary to bulk free fields with perturbative tree-level interactions providing the first deviation in $N^{-1}$ to factorized correlators. The Witten diagrams of these tree-level exchanges provide a useful graphical organization of the combinatorics of higher-point connected correlators. Because these are tree-level exchanges, from the bulk point of view, the connected correlators in Section 5.1 are not such a drastic extension: the bulk physics can still remain classical.[8] Here we use this fact to extend the regime of validity of our multi-trace states beyond full factorization to the regime of a weakly interacting bulk theory.

We consider states of the same form used above, defined in equation (80). As above, we assume that we have a scalar operator, $\mathcal{O}_\Delta$, in the theory with low enough dimension, $\Delta < d$, to allow for negative energy. When $\mathcal{O}_\Delta$ is single-trace in the large $N$ sense, this corresponds, from the bulk point of view, to a weakly interacting, BF-allowed, tachyon. Additionally, its corresponding multi-trace operator, $[\mathcal{O}_\Delta^p]$, has the bulk interpretation of a multi-particle operator

---

[8]However one still needs to specify the form of the bulk interactions. The corresponding CFT statement is specifying which single-trace operators can appear in single-trace OPEs.

formed from $p$ tachyons [30]. We now want to estimate the null energy in such a state. We have, schematically

$$\langle T_{--} \rangle_{\psi} = \frac{\langle [\mathcal{O}^p] T_{--} [\mathcal{O}^p] \rangle}{\langle [\mathcal{O}^p][\mathcal{O}^p] \rangle} \,. \tag{105}$$

To make the energy as negative as possible, we want to make $p$ large while maintaining computational control.

Full large $N$ factorization would require that the above correlators are approximated by the free theory in the bulk, with just one external graviton line and one interaction vertex. From the bulk point of view, we are inserting $p$ light particles. Assume that the smearing functions are chosen such that the wavelength of these particles is of order the AdS radius, $\ell$.

Let us compare the energy scale, $E_{\text{free}}$, of $p$ freely propagating particles of wavelength $\ell$ to the energy scale of gravitational interactions, $E_{\text{grav}}$. The free computation gives

$$E_{\text{free}} = \frac{p}{\ell} \,, \tag{106}$$

while an estimate for the energy in gravitational interactions is

$$E_{\text{grav}} = GM^2/r^{D-3} \sim \frac{G_N p^2}{\ell^{D-1}} \,, \tag{107}$$

where $D = d + 1$ is the bulk AdS spacetime dimension.

Combining these formulas tells us that the gravitational interactions are a small correction as long as

$$p \ll \frac{G_N}{\ell^{D-2}} \sim C_T \,. \tag{108}$$

This tells us that we can compute perturbatively in the bulk as long as the number of insertions $p$ is small compared to the central charge; it is allowed that $p$ scales as a small number times the central charge

$$p = \epsilon C_T \,. \tag{109}$$

Note that this is a weaker requirement than full large $N$ factorization. Full factorization would require that no gravitons are exchanged. We can estimate the number of gravitons exchanged from the gravitational interaction energy if we assume that the gravitons have wavelength $\ell$, giving

$$N_{\text{grav}} \sim \frac{G_N p^2}{\ell^{D-2}} \,. \tag{110}$$

Therefore, the thresholds are

$$\text{Free:} \quad p \ll \sqrt{C_T}\,, \qquad \text{Weak interactions:} \quad p \ll C_T\,. \tag{111}$$

In the regime of weak interactions, the number of gravitons is much smaller than the number $p$ of light scalars,

$$\frac{N_{\text{grav}}}{p} \sim \frac{p}{C_T} \ll 1\,. \tag{112}$$

Therefore, a typical light scalar does not attach to any graviton. As a result, we have

$$\langle [\mathcal{O}^p][\mathcal{O}^p] \rangle \sim p \langle \mathcal{O}\mathcal{O} \rangle \langle [\mathcal{O}^{p-1}][\mathcal{O}^{p-1}] \rangle \,. \tag{113}$$

This formula can be justified as follows: since most of the light scalars do not interact, the first $\mathcal{O}$ will, with high probability, connect to one of the $\mathcal{O}$ in the ket. There are $p$ choices for which one to attach to, and when this happens the amplitude takes the factorized form written

above. Note that this result is implied by full large $N$ factorization, but it is weaker, because we can be in the regime where the number of gravitons exchanged is large, so that $\langle[\mathcal{O}^p][\mathcal{O}^p]\rangle$ does not fully factorize into 2-point functions.

We need a similar result to evaluate the numerator. There, we make a similar argument: the external graviton corresponding to the stress-energy tensor attaches to one $\mathcal{O}$ from the bra and one from the ket with a cubic vertex. It has $p^2$ choices in which particles to attach to. By the same argument, this scalar line probably does not attach to any other gravitons, so in the weak interaction regime

$$\langle[\mathcal{O}^p]T_{--}[\mathcal{O}^p]\rangle \sim p^2 \langle\mathcal{O}T_{--}\mathcal{O}\rangle\langle[\mathcal{O}^{p-1}][\mathcal{O}^{p-1}]\rangle. \tag{114}$$

Again, this result is implied when full large $N$ factorization is valid but extends throughout the weak interaction regime $p \ll C_T$.

Combining the above formulas gives

$$\langle T_{--}\rangle_{\mathcal{O}^p} = \frac{\langle[\mathcal{O}^p]T_{--}[\mathcal{O}^p]\rangle}{\langle[\mathcal{O}^p][\mathcal{O}^p]\rangle} \sim \frac{p^2\langle\mathcal{O}T_{--}\mathcal{O}\rangle\langle[\mathcal{O}^{p-1}][\mathcal{O}^{p-1}]\rangle}{p\langle\mathcal{O}\mathcal{O}\rangle\langle[\mathcal{O}^{p-1}][\mathcal{O}^{p-1}]\rangle}. \tag{115}$$

The final result is

$$\langle T_{--}\rangle_{\mathcal{O}^p} \sim p\langle T_{--}\rangle_{\mathcal{O}^p}, \quad \text{for} \quad p \ll C_T. \tag{116}$$

This indicates that the negative null energy can scale as at least a small constant times the central charge.

## 6 Negative null energy in multi-stress tensor states

We now turn our attention to states prepared by stress-energy tensors. Firstly, as mentioned before, since the stress-energy tensor exists as an operator in every CFT this discussion lends some universality to its arguments. Secondly, as we will soon see, the ability to superpose stress-tensor states as in Section 3 will allow us to find negative null-energies with scaling greater than $C_T^{1/2}$.

In large $N$ theories, the stress-energy tensor is a single-trace operator, however we will be careful to note its special scaling conventions. By definition, the stress-energy tensor two-point function is proportional to the central charge of the CFT which sets the scaling of the two-point function to $N^2$. This also sets the scaling of the three-point and connected four-point functions

$$\langle TT\rangle \sim N^2, \qquad \langle TTT\rangle \sim N^2, \qquad \langle TTTT\rangle_{\text{conn}} \sim N^2, \tag{117}$$

where we have dropped indices and function dependence instead focussing on the overall $N$-scaling and again the connected four-point function is given by

$$\langle T_1T_2T_3T_4\rangle_{\text{conn}} := \langle T_1T_2T_3T_4\rangle - \langle T_1T_2\rangle\langle T_3T_4\rangle - \langle T_1T_3\rangle\langle T_2T_4\rangle - \langle T_1T_4\rangle\langle T_2T_3\rangle. \tag{118}$$

This scaling continues to apply to all connected higher-point functions which implies that $n$-point functions are dominated by their disconnected constituents. As in the multi-trace operators formed from the subsequent OPE of single-trace scalar operators, it is easy to see how factorization implies the existence of a double-trace operators appearing in the of $T_{ab}$ with itself. We will focus on the lowest twist spin-4 double-trace operator, which we will call the *double-stress tensor*. This can be thought of as a normal-ordered product of $T_{ab}$:

$$[T^2]_{a_1,b_1;a_2,b_2} \sim \ :T_{a_1b_1}T_{a_2b_2}:\ . \tag{119}$$

This operator has conformal dimension $\Delta_T^{(2)} = 2d + \gamma_T^{(2)}$ where we can argue that $\gamma_T^{(2)} = O(N^{-2})$. While the operator product of generic components of the stress-energy tensor in a typical configuration is certainly more involved due to the compounding of tensor factors, the arguments given for scalar operators in the previous subsection apply straightforwardly to stress-energy tensor OPEs of $T_{\tau\tau}$ arranged collinearly in Euclidean time.

In a similar fashion to the previous subsection, this construction can be iterated to build a series of multi-trace operators through the successive OPE with $T_{ab}$. In this paper we will call the particular multi-trace primary appearing in the $p^{\text{th}}$ OPE of $T_{ab}$ with lowest twist and spin $2p$ the $p^{\text{th}}$ *multi-stress tensor*, $[T^p]_{\{a_i, b_i\}}$ where $\{a_i, b_i\} = a_1, b_1; a_2, b_2; \ldots; a_p, b_p$ is shorthand for a multi-index that is symmetric and traceless on any pair $(a_i, b_i)$ and symmetric under the interchange $(a_i, b_i) \leftrightarrow (a_j, b_j)$. Much like the muti-trace operators utilized in the Section 4, this definition does not specify $[T^p]$ uniquely, say due to mixing with single-trace operators, but only at leading order in $N$.[9] For our present purposes this is enough, although below we will discuss when $1/N$ corrections become important.

## 6.1 States with negative null energy

Just as in the previous section we can construct a multi-stress tensor state by aggregating integrated stress-energy tensors in their OPE. That is, given a Euclidean polarization function $h^{ab}(x)$ we can consider the state

$$|T_h^p\rangle = \mathcal{N} (T_h)^p |\Omega\rangle, \tag{120}$$

with

$$(T_h)^p \equiv \int \left( \prod_{i=1}^{p} d^d x_i\, h^{a_i b_i}(x_i) \right) \left( \prod_{i=1}^{p-1} \lambda_{\Delta_T \Delta_T^{(i)} \Delta_T^{(i+1)}} \right) (\mathcal{P} \cdot \mathcal{P} \cdot \ldots \cdot \mathcal{P})_{\{a_i, b_i\}}^{\{c_i, d_i\}} [T^p]_{\{c_i, d_i\}}(x_1), \tag{121}$$

where

$$(\mathcal{P} \cdot \mathcal{P} \cdot \ldots \cdot \mathcal{P})_{\{a_i, b_i\}}^{\{c_i, d_i\}} \equiv \mathcal{P}_{a_p, b_p\ \ A_{p-1}}^{\{c_i, d_i\}}(x_{p, p-1}; \partial^{(x_{p-1})}) \ldots \mathcal{P}_{a_3, b_3\ \ A_2}^{A_3}(x_{32}; \partial^{(x_2)})$$
$$\times \mathcal{P}_{a_2, b_2\ \ a_1, b_1}^{A_2}(x_{21}; \partial^{(x_1)}), \tag{122}$$

and $A_i$ is a multi-index running over the irreducible $SO(d)$ representations appering in the $i^{\text{th}}$ tensor product of the traceless-symmetric spin-two representation. This cumbersome notation only formalizes a very simple statement: correlation functions of $(T_h)^p$ factorize, at leading order in $N$, into two-point functions of $\int d^d x\, h^{ab} T_{ab}$. The arguments of the previous subsection apply here and the null energy density in this state is given, to leading order in $N$, by

$$\int d^d x\, g(x)^2 \langle T_{--}(x) \rangle_{T_h^p} = p \frac{\int d^d x\, g(x)^2 \langle T_h^\dagger T_{--}(x) T_h \rangle}{\langle T_h^\dagger T_h \rangle} + O(1/N). \tag{123}$$

The sign of this expectation value is then set by the sign of the integrated stress-energy tensor three-point function. Whether or not this three-point function can be made negative or not, generically, is still open. In Section 3 we have presented evidence that in $d \geq 4$ dimensions the pointwise expectation values are positive. While this obviously does not imply the positivity of an integrated three-point function, we have, as of yet, been unable to construct one. In $d = 3$ dimensions we have found a negative pointwise expectation value in a colinear configuration for a particular polarization. In this case, by taking $h^{ab}$ to be aligned with this polarization and strongly localized to a colinear configuration, it seems possible that the integrated expectation

---

[9]We thank Nat Levine for discussions on this point.

value could be negative. Regardless, at least for $p \sim O(N^0)$ operators then this negative energy density also remains order $N^0$.

However, we can try a different strategy for generating negative null energy densities, regardless of the sign of the three-point function. We do this by considering superpositions:

$$|\alpha_p\rangle := \mathcal{N}_p \left( |T_h^p\rangle + \alpha_p |T_h^{p+1}\rangle \right), \tag{124}$$

where $\mathcal{N}_p$ normalizes the state. For brevity of notation, let us denote the following

$$T[g] := \int d^d x \, g(x)^2 T_{--}(x). \tag{125}$$

To leading order in $N$ the smeared null energy is given by

$$\langle T[g]\rangle_{\alpha_p} \approx \mathcal{N}_p^2 \Big\{ p^2(p-1)! \langle T_h^\dagger T[g] T_h\rangle \langle T_h^\dagger T_h\rangle^{p-1} + 2(p+1)p! \, \mathrm{Re}\left( \alpha_p \langle T[g] T_h\rangle \right) \langle T_h^\dagger T_h\rangle^p$$
$$+ (p+1)^2 p! \left| \alpha_p \right|^2 \langle T_h^\dagger T[g] T_h\rangle \langle T_h^\dagger T_h\rangle^p \Big\}, \tag{126}$$

with

$$\mathcal{N}^2 \approx \left\{ p! \langle T_h^\dagger T_h\rangle^p + (p+1)! \left| \alpha_p \right|^2 \langle T_h^\dagger T_h\rangle^{p+1} \right\}^{-1}, \tag{127}$$

at leading order in $N$. The benefit of this superposition is that we are free to choose the sign of $\alpha_p$ and we can use this freedom to make this expectation value negative. Let us choose the phase of $\alpha_p$ such that

$$\alpha_p \langle T[g] T_h\rangle = -\left| \alpha_p \right| \left| \langle T[g] T_h\rangle \right|, \tag{128}$$

and minimize over $\left| \alpha_p \right|$. The minimizer, $\bar{\alpha}_p$, is easy to find and given by

$$\bar{\alpha}_p = \frac{-\langle T_h^\dagger T[g] T_h\rangle + \sqrt{\left( \langle T_h^\dagger T[g] T_h\rangle \right)^2 + 4(p+1)\langle T_h^\dagger T_h\rangle |\langle T[g] T_h\rangle|^2}}{2(p+1)\langle T_h^\dagger T_h\rangle}, \tag{129}$$

and the minimum is given by

$$\langle T[g]\rangle_{\bar{\alpha}_p} = (2p+1)\frac{\langle T_h^\dagger T[g] T_h\rangle}{2\langle T_h^\dagger T_h\rangle} - \frac{\sqrt{\left( \langle T_h^\dagger T[g] T_h\rangle \right)^2 + 4(p+1)\langle T_h^\dagger T_h\rangle |\langle T[g] T_h\rangle|^2}}{2\langle T_h^\dagger T_h\rangle}. \tag{130}$$

It is clear that this expectation value can be negative. For instance, for $p \sim O(N^0)$ this expectation value scales as

$$\langle T[g]\rangle_{\bar{\alpha}_p} \sim -N \sim -C_T^{1/2}. \tag{131}$$

## 6.2 How negative can it be?

In the previous subsection, we illustrated how the possibility of generating negative smeared null energy in multi-stress tensor states. We illustrated this both in states prepared by integrated operators as well as in superposition states. In either case, the smeared null energy could be made more negative by increasing the order, $p$, of the multi-stress tensor operators involved. This might suggest that the smeared null energy is unbounded below. This would possibly be the case if we could evaluate $n$-point functions utilizing exact Wick contractions, as in free-field theory. However, since we rely on large $N$ factorization, there is only a limited range in $p$ for which our estimations apply.

In particular, it is easy to generalize the arguments of subsection 4.3 to multi-stress tensor operators to see that two-point functions dominate over three-point functions and connected four-point functions in the computation of $\langle [T^p][T^p]\rangle$ when $p$ scales with $N$ at most like

$$p_{\max} \sim N \,. \tag{132}$$

In this maximal scaling the smeared null energy in integrated multi-stress tensor states could possibly scale as

$$\int d^d x\, g(x)^2 \langle T_{--}(x)\rangle_{h^{p_{\max}}} \sim -N \sim -C_T^{1/2}\,, \tag{133}$$

while remaining in the regime where large $N$ factorization remains valid. In our superposition states, this negative null energy is slightly more drastic, scaling as

$$\int d^d x\, g(x)^2 \langle T_{--}(x)\rangle_{\tilde{\alpha}_{p_{\max}}} \sim -N^{3/2} \sim -C_T^{3/4}\,. \tag{134}$$

## 6.3   Beyond $C_T^{3/4}$

As in the discussion above, we expect that the null energy can be more negative in general. We can repeat our analysis above to go beyond order $C_T^{3/4}$ negative smeared null energy. Including contributions from connected correlators is more involved in this case, and we do not carry out the analysis explicitly, but we expect the results would be similar. On the other hand, the arguments for product CFT's go through exactly as in the scalar case.

We will do the analysis more explicitly in holographic theories. Consider states of the form

$$|\psi\rangle = T^p\,|\Omega\rangle + \alpha T^{p+1}\,|\Omega\rangle \,, \tag{135}$$

where $T^p$ is taken to be normal ordered, as above. We now want to estimate the null energy in such a state. We have

$$\langle T_{--}\rangle_\psi = \frac{\langle T^p T_{--} T^p\rangle + \alpha^* \langle T^{p+1} T_{--} T^p\rangle + c.c. + |\alpha|^2 \langle T^{p+1} T_{--} T^{p+1}\rangle}{\langle T^p T^p\rangle + |\alpha|^2 \langle T^{p+1} T^{p+1}\rangle}\,. \tag{136}$$

We want to diagnose how large $p$ can be before large $N$.

Look first at the norm of the state. We want to impose something milder than full large $N$ factorization. We want simply that

$$\langle T^{p+1} T^{p+1}\rangle = f(p)\,\langle T\,T\rangle\,\langle T^p T^p\rangle\,. \tag{137}$$

Consider all of the bulk diagrams connecting $p+1$ gravitons in the initial state to the same number of gravitons in the final state. We will use the same logic as in the scalar case above, so will be briefer in the explanations.

Suppose that a typical graviton in the initial state does not interact. In this case, we can calculate $\langle T^p T^p\rangle$ by first factoring out the first $T$. Since typical gravitons do not interact, this first $T$ will not interact, and has a choice of $p$ final $T$'s to connect to. Therefore, in this regime,

$$\langle T^p T^p\rangle = \langle T\,T\rangle\langle T^{p-1} T^{p-1}\rangle\,. \tag{138}$$

The regime of validity of this approximation can be estimated in the same way as in the scalar case, and requires

$$p \ll C_T\,. \tag{139}$$

Similarly, we have

$$\langle T^p T_{--} T^p\rangle \sim p^2\,\langle T\,T_{--} T\rangle\langle T^{p-1} T^{p-1}\rangle\,. \tag{140}$$

We also need to evaluate

$$\left\langle T^p T^{p+1} \right\rangle = 0 \,, \tag{141}$$

due to the normal ordering prescription, which guarantees that $T^p$ and $T^{p+1}$ are distinct operators with different scaling dimension. Finally, we need

$$\left\langle T^p T_{--} T^{p+1} \right\rangle \sim (p+1) \left\langle T_{--} T \right\rangle \left\langle T^p T^p \right\rangle \,, \tag{142}$$

which follows from similar logic: the $T_{--}$ contracts with one of the $p+1$ $T$ operators in the ket.

With these results in hand, we can estimate

$$\langle T_{--} \rangle_\psi \sim \frac{\langle T T_{--} T \rangle \left( p^2 \langle T^{p-1} T^{p-1} \rangle + |\alpha|^2 (p+1)^2 \langle T^p T^p \rangle \right) + 2(p+1) \mathrm{Re}\left[ \alpha \langle T_{--} T \rangle \langle T^p T^p \rangle \right]}{\langle T^p T^p \rangle + |\alpha|^2 (p+1) \langle T T \rangle \langle T^p T^p \rangle} \,. \tag{143}$$

Since $p$ is large we will not distinguish between $p$ and $p+1$ in the numerical prefactors. Simplifying the above expression gives

$$\langle T_{--} \rangle \sim \frac{p \langle T T_{--} T \rangle / \langle T T \rangle + p \, \mathrm{Re}[\alpha \langle T_{--} T \rangle] + |\alpha|^2 p^2 \langle T T_{--} T \rangle}{1 + |\alpha|^2 p \langle T T \rangle} \,. \tag{144}$$

We can find the optimal value of $\alpha$, as above, but here we content ourselves with determining the scaling with central charge. We use the normalization that connected multi-point correlators of the stress-energy tensor are proportional to $C_T$. We then have

$$\langle T_{--} \rangle \sim \frac{p + \mathrm{Re}[\alpha] p C_T + |\alpha|^2 p^2 C_T}{1 + |\alpha|^2 p C_T} \,. \tag{145}$$

The optimization is more transparent if we define $\alpha \sqrt{p C_T} = -\gamma$. With this sign convention it is clear we want $\gamma$ real and positive to minimize:

$$\langle T_{--} \rangle \sim \frac{p - \gamma \sqrt{p C_T} + \gamma^2 p}{1 + \gamma^2} \sim p \frac{1 - \gamma \sqrt{\frac{C_T}{p}} + \gamma^2}{1 + \gamma^2} \,. \tag{146}$$

To remain in the weak interaction regime, $\sqrt{\frac{C_T}{p}}$ must be large. In this regime this is the only term that matters in the numerator, and the optimal value of $\gamma$ is $\gamma = 1$, with

$$\langle T_{--} \rangle \sim -\sqrt{p \, C_T} \,. \tag{147}$$

As before, we require only that $p \ll C_T$ for our approximation to work, so we can construct states the negative energy that scales linearly in $C_T$.

A nice aspect of the states constructed here is that some holographic theories may not have sufficiently light scalars in the spectrum for the construction in the previous section, but all holographic theories have gravity.

## 7 Discussion

In this paper we have investigated integrated null energy densities in large $N$ interacting CFTs. We showed, firstly, that scalar operators of sufficiently heavy conformal dimension, $\Delta \geq d$, prepare states with positive null energy densities. However, if the theory has light scalar operators with $\Delta < d$, then one can prepare states with negative smeared null energy. Secondly, we showed that states prepared by the multi-trace operators appearing in the OPE of light scalar operators can lead to smeared null energies scaling with the order of the multi-trace

operator. Unlike free field theory, however, this does not imply that the smeared null energy is unbounded: namely, insisting upon a strict factorization of correlation functions sets an upper limit on the order of the multi-trace operators used in preparing states. Within this class of states, we showed that the smeared null energy can scale at worst as $\sqrt{C_T}$. We gave additional arguments for states in which this scaling could be pushed to $\sim C_T$. The result from two-dimensions, (7), and our arguments from product CFTs in Section 5 lead us to suspect this is the maximal scaling of the smeared null energy, however we have not proved so in this paper.

While the above results rely on the existence of a sufficiently light single-trace operator, we illustrated that many of the same conclusions can be reached in states prepared by integrated stress-energy tensors (which exist in every CFT). Namely, by taking superpositions of multi-stress tensor operators appearing in the OPE of multiple $T_{ab}$, we showed that one can build states with negative smeared null energy scaling as $C_T^{3/4}$ while preserving factorization of correlation functions. Along the way, we presented arguments for the positivity of stress-tensor three-point functions in dimensions $d \geq 4$, which may be a result of independent interest. Lastly, we gave further arguments towards pushing the maximal scaling of the negative smeared null energy to $\sim C_T$. Again, we leave the question of whether this scaling can be pushed past $C_T$ to further work (although again we suspect not).

Our work suggests that some of the pathologies found in free field theory (unbounded negative null energy densities in particular) are corrected by the role of interactions. In our introduction, Section 1, we formalized and extended this suggestion in the form of two conjectures. The strong form of our conjecture is that all interacting CFTs have smeared null energies lower bounded by a state-independent functional which is linear in the central charge(s) of the theory. In its weak form, our conjecture allows a potential lower bound to have a linear state-dependence through the expectation value of a bounded number of light primary operators. These expectation values would provide the parameters defining an effective field theory of states with lower bounded null energy densities.

In addition to proving one of our the above conjectures for generic interacting CFTs, there are additional open directions that are naturally implied by our work. Let us briefly discuss them below.

## Connections to QNEC

Following up on the above discussion, one lower bound on null energy densities that has been proven for both free [43] and interacting CFTs [44] is the *quantum null energy condition* (QNEC) [45]. The QNEC lower bounds the pointwise expectation value of the null energy by the second null variation of a state's entanglement entropy with respect to its entangling surface. Moreover, under mild assumptions, the QNEC is saturated for generic interacting theories [46,47]. Despite this, the QNEC faces some shortcomings: both sides of inequality can diverge leaving the bound (and its saturation) indeterminate. Additionally, the entanglement entropy is a highly non-linear function of the state. This makes it difficult to compute and to utilize in semi-classical singularity theorems (although see [48,49] for progress).

Deriving a lower bound on the smeared null energy density with a milder (linear) state dependence directly from the QNEC would be incredibly interesting.[10] It seems unlikely that directly integrating the QNEC will provide any leverage towards this end, however there is promise in modifying the proof of QNEC through the OPE of displacement operators [47] to yield a lower bound on the smeared stress-energy tensor. Connecting QNEC (and its saturation) to a general lower bound on smeared null energy is an important future research direction.

---

[10]We thank Andrew Rolph for discussions along these lines.

**Further connections to holography**

While we have not relied explicitly on AdS/CFT for many of the results of this paper, we have used holographic intuition to argue for the existence of states with negative smeared null energy proportional to $C_T$. In Sections 5.3 and 6.3 we argued such states are prepared by multi-trace operators of order $p \sim C_T \sim N^2$ while remaining in a regime of a classical, weakly interacting, bulk physics. Quantum bulk physics becomes important (i.e. loops appearing in Witten diagrams become combinatorically favourable) only when computing correlators of order $p \gtrsim N^2$. The reader might be surprised that we can construct CFT states with negative smeared null energy scaling close to $C_T$ while the bulk physics remains classical and, importantly, NEC obeying. There is no strict contradiction here, *per se*, because, under the holographic dictionary, the CFT stress-energy tensor is related to the boundary metric fluctuation as opposed to the bulk energy operator. These are, of course, related via the bulk Einstein equation when the bulk remains classical. It is intriguing to speculate on connections between bulk classical energy conditions and CFT QEIs. One such connection is the proof of the *smeared null energy condition* [18, 21] for holographic CFTs coupled to dynamical gravity induced on a brane in AdS/CFT, by utilizing the classical "no bulk shortcut" principle [50]. More generally, a more direct connection between bulk energy conditions (either classical or quantum) and CFT QEIs would be a useful data point for the existence of QEIs in generic CFTs.

**Connections to singularity theorems and wormhole spacetimes**

As mentioned in the introduction, the DSNEC with one of the lengths held constant, can be used to prove the Penrose singularity theorem in semiclassical gravity [18, 24]. If one of our conjectures for CFTs is proven, the results could be applied to derive a singularity theorem for CFTs. It would be an important generalization of the current work to examine under what conditions of the CFT a singularity can be formed in the corresponding classical spacetime. A different research direction would be to use bounds of integrals on more than one spacetime direction for singularity theorems such as the DNEC. Progress has been made in the timelike case [51] with a version of Hawking's singularity theorem with a worldvolume energy bound. DSNEC can also be applied to long wormholes where the condition can be used on achronal segments inside the wormhole throat [19] and determine further restrictions to the possibility of the existence of a long wormhole.

# Acknowledgments

We thank Agnese Bissi, Alejandra Castro, Diego Hofman, Gerardo García-Moreno, Avner Karasik, Nat Levine, Rishi Mouland, Andrew Rolph, Kamran Salehi Vaziri, Aron Wall, Bernardo Zan, and Sasha Zhiboedov for helpful conversations. JRF thanks the University of Amsterdam, King's College London, and CERN for hospitality during the completion of this work.

**Funding information** JRF is partially supported by STFC consolidated grants ST/T000694/1 and ST/X000664/1, and by Simons Investigator award #620869. DPS is supported by the EPSRC studentship grant EP/W524475/1. BF is partially supported by Heising-Simons Foundation 'Observational Signatures of Quantum Gravity' QuRIOS collaboration grant 2021-2817.

# A Positivity of state norms

In this appendix, we show that certain integrals arising in the main body are positive. These integrals arose in, for example, equation (33) and are of the form

$$\int_{H_-} d^d x_1 d^d x_2 h(x_1)^* h(x_2) \langle \mathcal{O}_\Delta(x_1)^\dagger \mathcal{O}_\Delta(x_2) \rangle = \int_{H_-} d^d x_1 d^d x_2 \frac{h(x_1)^* h(x_2)}{\left| x_1^R - x_2 \right|^{2\Delta}} . \qquad (A.1)$$

The quick argument is that this has the form of the norm of a state, so must be positive for dimensions satisfying the unitarity bound

$$\Delta \geq \frac{d-2}{2} . \qquad (A.2)$$

However, we want to show this explicitly since not all integrals of this form arise from a norm, so we want to be sure there are no hidden assumptions about the state preparation functions $h$.

To show this, we begin with a Fourier transform result

$$\frac{1}{r^\alpha} = c_{\alpha n} \int \frac{d^n K}{(2\pi)^n} e^{iK_a x^a} \frac{1}{|K|^{n-\alpha}} , \qquad (A.3)$$

valid for $0 \leq \alpha \leq n$. Here $c_{\alpha n}$ is a positive constant.

Note that we need to deal with arbitrarily large powers $\alpha$ in a fixed number of dimensions. To deal with this, we allow the number of dimensions $n$ in the Fourier transform to be larger than the actual spacetime dimension $d$. Because our smearing function treats space and Euclidean time differently, it will be useful to distinguish these. We will also distinguish between the physical spacetime dimensions $(\omega, \vec{k})$ and the auxilliary dimensions, which we call $\vec{p}$. We have

$$\frac{1}{r^\alpha} = c_{\alpha n} \int \frac{d\omega d^{d-1}k d^{n-d}p}{(2\pi)^n} e^{i(\omega\tau + \vec{k}\cdot\vec{x})} \frac{1}{|k^2 + p^2 + \omega^2|^{\frac{n-\alpha}{2}}} . \qquad (A.4)$$

The physical intuition is that we obtain high powers by embedding the physical dimensions in a higher dimensional space. We can think of the physical plane as sitting at the origin in the extra dimensions.

We now want to deform the contour of the $\omega$ integral, assuming $\tau > 0$. In the complex $\omega$ plane, we have branch points at $\omega = \pm i\sqrt{k^2 + p^2}$; we take the branch cuts along the imaginary axis as shown in the figure.

After deforming the contour as shown in Figure 3, the integral becomes

$$\frac{1}{r^\alpha} = 2c_{\alpha n} \sin\left(\frac{\pi(n-\alpha)}{2}\right) \int \frac{d^{d-1}k d^{n-d}p}{(2\pi)^n} \int_{\sqrt{k^2+p^2}}^\infty dw e^{i\vec{k}\cdot\vec{x} - w\tau} \frac{1}{|w^2 - (k^2 + p^2)|^{\frac{n-\alpha}{2}}} , \qquad (A.5)$$

where the sin arises from the difference in the integrand on the two sides of the branch cut.

Note that this formula is useful if the $w$ integral is well-behaved near its lower limit, requiring

$$\alpha > n - 2 . \qquad (A.6)$$

In this regime the prefactor is positive. We now want to change the order of limits and do the integral over the auxilliary $p$ coordinates, obtaining

$$\frac{1}{r^\alpha} = \tilde{c}_{\alpha n} \int d^{d-1}k \int_{|k|}^\infty dw e^{ik\cdot x - w\tau} \int_0^{\sqrt{w^2 - k^2}} d^{n-d}p \frac{1}{|w^2 - k^2 - p^2|^{\frac{n-\alpha}{2}}} . \qquad (A.7)$$

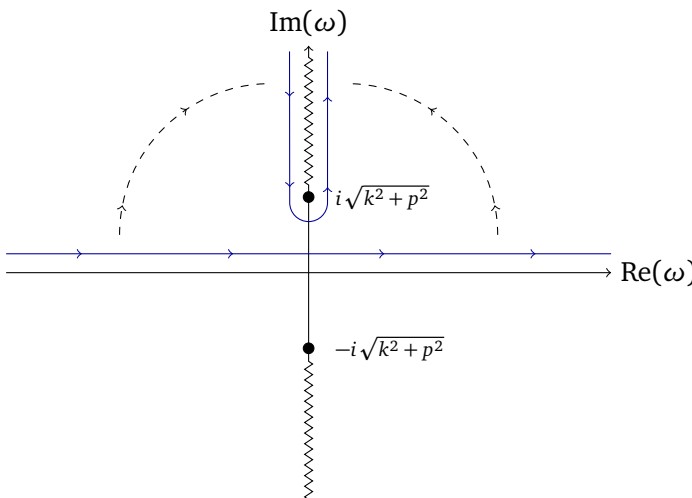

Figure 3: The contour deformation yielding the integral (A.5).

The $p$ integral converges in the range of alpha of interest (A.6) and, up to the overall positive constant, can be done by dimensional analysis, yielding

$$\frac{1}{r^\alpha} = C_{\alpha n d} \int d^{d-1}k \int_{|k|}^{\infty} dw e^{ik\cdot x - w\tau}(w^2 - k^2)^{(\alpha - d)/2}. \tag{A.8}$$

At this point, the auxilliary coordinates have been integrated out. The choice of $n$ does not matter except to make integrals converge in the intermediate steps, so in fact the overall positive constant should not depend on $n$ and we have derived a simple result:

$$\frac{1}{r^\alpha} = C_{\alpha d} \int d^{d-1}k \int_{|k|}^{\infty} dw e^{ik\cdot x - w\tau}(w^2 - k^2)^{(\alpha - d)/2}, \qquad \text{for} \quad \alpha > d - 2, \tag{A.9}$$

where $C_{\alpha d}$ is a positive constant.

When the inequality is saturated, $\alpha = d - 2$, a similar formula holds,

$$\frac{1}{r^{d-2}} = C_d \int d^{d-1}k \int_0^{\infty} dw e^{ik\cdot x - w\tau}\delta(w^2 - k^2). \tag{A.10}$$

Presumably these formulas have been derived many times before, but we were not able to find a useful reference. The result is simply a Fourier-Laplace transform of the power law.

With these formulas in hand, positivity follows quickly. Returning to the integral of interest,

$$I = \int d\tau_1 d^{d-1}x_1 d\tau_2 d^{d-1}x_2 h^*(\tau_1, \vec{x}_1) h(\tau_2, \vec{x}_2)\frac{1}{[(\tau_1 + \tau_2)^2 + (\vec{x}_1 - \vec{x}_2)^2]^\Delta}, \tag{A.11}$$

where the preparation function $h$ only has support for positive $t$. Making use of the representation of the power law derived above gives

$$\begin{aligned} I = &C_{\Delta d} \int d\tau_1 d^{d-1}x_1 d\tau_2 d^{d-1}x_2 \int d^{d-1}k \\ &\times \int_{|k|}^{\infty} dw h^*(\tau_1, \vec{x}_1) h(\tau_2, \vec{x}_2) e^{ik\cdot(x_1 - x_2) - w(\tau_1 + \tau_2)}(w^2 - k^2)^{\Delta - d/2}. \end{aligned} \tag{A.12}$$

We now rearrange the integrals to obtain

$$
\begin{aligned}
I = C_{\Delta d} \int d^{d-1}k \int_{|k|}^{\infty} dw (w^2 - k^2)^{\Delta - d/2} \int d\tau_1 d^{d-1}x_1 h^*(x_1, \tau_1) e^{ik \cdot x_1 - w\tau_1} \\
\times \int d\tau_2 d^{d-1}x_2 h(x_2, t_2) e^{-ik \cdot x_2 - w\tau_2}.
\end{aligned}
\tag{A.13}
$$

This is now the integrand of a perfect square:

$$
I = C_{\Delta d} \int d^{d-1}k \int_{|k|}^{\infty} dw (w^2 - k^2)^{\Delta - d/2} \left| \int d\tau d^{d-1}x \, h(x, \tau) \, e^{-ik \cdot x - wt} \right|^2.
\tag{A.14}
$$

The quantity appearing inside the absolute value is just the Fourier-Laplace transform of the state preparation function $h$, and this formula demonstrates that the integral is just a particular norm of this function.

The result can be extended to the edge case $\Delta = (d-2)/2$ using similar arguments. Therefore, we have shown that the integral of interest is non-negative when $\Delta$ satisfies the unitarity bound, $\Delta \geq (d-2)/2$.

# B  Two and three-point functions in CFTs

This appendix summarises established results used in the main text for two- and three-point functions of the stress-energy tensor and scalar operators for unitary CFTs in $d \geq 3$, along with constraints on their coefficients derived from ANEC.

## B.1  The two-point function

Imposing conformal symmetry and the conservation of the stress-energy tensor, the two-point function of $T_{ab}$ has the following structure [28, 52]

$$
\langle T_{ab}(x) T_{cd}(0) \rangle = \frac{C_T}{x^{2d}} \mathcal{I}_{ab,cd}(x),
\tag{B.1}
$$

where $C_T$ characterizes the leading singularity of the two-point function and[11]

$$
\mathcal{I}_{ab,cd}(s) = I_{ae}(s) I_{bf}(s) \mathcal{E}_{ef,cd},
\tag{B.2}
$$

with $I_{ab}(x) = \delta_{ab} - 2\frac{x_a x_b}{x^2}$ the inversion operator. $\mathcal{E}$ is given by

$$
\mathcal{E}_{ab,cd} = \frac{1}{2}(\delta_{ac}\delta_{bc} + \delta_{ad}\delta_{bc}) - \frac{1}{d}\delta_{ab}\delta_{cd},
\tag{B.3}
$$

and it is the projection operator onto the space of symmetric traceless tensors. $\mathcal{I}$ represents the inversion tensor associated to $\mathcal{E}$.

From the positivity of the two-point function in unitary theories, it is necessary that $C_T > 0$. In even dimensions, $C_T$ is related to the coefficient of the trace anomaly of the CFT.

---

[11]Note that the Einstein summation convention is used here.

## B.2 The three-point function

### B.2.1 The general frame

The general expression for the three-point function of the stress-energy tensor with two scalar operators $\mathcal{O}$ of conformal dimension $\Delta$ is [28]

$$\langle T_{ab}(x_1)\mathcal{O}(x_2)\mathcal{O}(x_3)\rangle = \frac{C_{T\mathcal{O}\mathcal{O}}}{x_{12}^d x_{13}^d x_{23}^{2\Delta-d}} t_{ab}(X), \tag{B.4}$$

where

$$t_{ab}(X) = \frac{X_a X_b}{X^2} - \frac{1}{d}\delta_{ab}, \qquad X = \frac{x_{12}}{x_{12}^2} - \frac{x_{13}}{x_{13}^2}, \tag{B.5}$$

and $x_{ij} = x_i - x_j$. The coefficient is given by $C_{T\mathcal{O}\mathcal{O}} = -\frac{d}{d-1}\frac{\Delta}{\Omega_{d-1}}$ where $\Omega_{d-1} = \frac{2\pi^2}{\Gamma(d/2)}$ is the area of the unit $(d-1)$-sphere [28].

Similarly, conformal symmetry leads to a simplified expression for the three-point functions of the stress-energy tensor at generic points in the spacetime [28]

$$\langle T_{ab}(x_1)T_{cd}(x_2)T_{ef}(x_3)\rangle = \frac{1}{x_{12}^d x_{13}^d x_{23}^d}\mathcal{I}_{ab,a'b'}(x_{13})\mathcal{I}_{cd,c'd'}(x_{23})t_{a'b'c'd'ef}(X_{12}), \tag{B.6}$$

where $X_{12} = \frac{x_{13}}{x_{13}^2} - \frac{x_{23}}{x_{23}^2}$ and $t_{abcdef}(X_{12})$ is an homogeneous of degree zero in $X$ tensor, symmetric and traceless on each pair of indices $ab$, $cd$ and $ef$, that satisfies

$$t_{abcdef}(X) = t_{cdabef}(X), \tag{B.7}$$

$$\mathcal{I}_{ab,a'b'}(X)t_{a'b'cdef}(X) = t_{efabcd}(X). \tag{B.8}$$

A general expansion for $t_{abcdef}(X)$ compatible with the previous conditions is [28]

$$\begin{aligned}
t_{abcdef}(X) =\; & a h^5_{abcdef} + b h^4_{efabcd}(\hat{X}) + b'(h^4_{abcdef}(\hat{X}) + h^4_{cdabef}(\hat{X})) \\
& + c h^3_{abcd} h^1_{ef}(\hat{X}) + c'(h^3_{cdef} h^1_{ab}(\hat{X}) + h^3_{abef} h^1_{cd}(\hat{X})) \\
& + e h^2_{abcd}(\hat{X}) h^1_{ef}(\hat{X}) + e'(h^2_{cdef}(\hat{X}) h^1_{ab}(\hat{X}) + h^2_{abef}(\hat{X}) h^1_{cd}(\hat{X})) \\
& + f h^1_{ab}(\hat{X}) h^1_{cd}(\hat{X}) h^1_{ef}(\hat{X}),
\end{aligned} \tag{B.9}$$

where

$$\begin{aligned}
h^1_{ab}(\hat{X}) &= \hat{X}_a \hat{X}_b - \frac{1}{d}\delta_{ab}, \quad \text{with} \quad \hat{X}_a = \frac{X_a}{\sqrt{X^2}}, \\
h^2_{abcd}(\hat{X}) &= \hat{X}_a \hat{X}_b \delta_{bd} + (a \leftrightarrow b, c \leftrightarrow d) - \frac{4}{d}\hat{X}_a \hat{X}_b \delta_{cd} - \frac{4}{d}\hat{X}_c \hat{X}_d \delta_{ab} + \frac{4}{d^2}\delta_{ab}\delta_{cd}, \\
h^3_{abcd} &= \delta_{ac}\delta_{bd} + \delta_{ad}\delta_{bc} - \frac{2}{d}\delta_{ab}\delta_{cd}, \\
h^4_{abcdef}(\hat{X}) &= h^3_{abcd}\hat{X}_d \hat{X}_f + (c \leftrightarrow d, e \leftrightarrow f) - \frac{2}{d}\delta_{cd}h^2_{abef}(\hat{X}) - \frac{2}{d}\delta_{ef}h^2_{abcd}(\hat{X}) \\
& \quad - \frac{8}{d^2}\delta_{cd}\delta_{ef}h^1_{ab}(\hat{X}), \\
h^5_{abcdef} &= \delta_{ac}\delta_{be}\delta_{df} + (a \leftrightarrow b, c \leftrightarrow d, e \leftrightarrow f) - \frac{4}{d}\delta_{ab}h^3_{cdef} - \frac{4}{d}\delta_{cd}h^3_{abef} \\
& \quad - \frac{4}{d}\delta_{ef}h^3_{abcd} - \frac{8}{d^2}\delta_{ab}\delta_{cd}\delta_{ef}.
\end{aligned} \tag{B.10}$$

Eq. (B.8) together with the conservation of the stress-energy tensor impose five relations among the eight coefficients, $(a, b, b', c, c', e, e', f)$, in (B.9). Therefore, the three-point

function of the stress-energy tensor of any CFT is fully characterized by three independent coefficients. In addition, the Ward identities relate the two- and the three-point functions [28, 52]

$$C_T = 4\Omega_{d-1} \frac{(d-2)(d+3)a - 2b - (d+1)c}{d(d+2)} \,. \tag{B.11}$$

Therefore, we can write the three-point functions of the stress-energy tensor in terms of two independent coefficients and $C_T$.

### B.2.2 The collinear frame

The expression for the three-point functions of the stress-energy tensor simplifies when the three spacetime points lie on a straight line [28]

$$\langle T_{ab}(x_1) T_{cd}(x_2) T_{ef}(x_3) \rangle = \frac{1}{|\tau_1 - \tau_2|^d |\tau_1 - \tau_3|^d |\tau_2 - \tau_3|^d} \mathcal{A}_{abcdef} \,, \tag{B.12}$$

where $x_\mu = (\tau, \mathbf{0})$, $\mathcal{A}_{abcdef} = \mathcal{A}_{cdabef} = \mathcal{A}_{efabcd}$, and $\mathcal{A}_{abcdef}$ is symmetric and traceless on each pair of indices. With these conditions, we have

$$
\begin{aligned}
\mathcal{A}_{\tau\tau\tau\tau\tau\tau} =& \alpha \,, \\
\mathcal{A}_{ij\tau\tau\tau\tau} =& \beta \delta_{ij} \,, \\
\mathcal{A}_{i\tau k\tau\tau\tau} =& \gamma \delta_{ik} \,, \\
\mathcal{A}_{ijkl\tau\tau} =& \delta \delta_{ij} \delta_{kl} + \epsilon(\delta_{ik}\delta_{jl} + \delta_{il}\delta_{jk}) \,, \\
\mathcal{A}_{ijk\tau m\tau} =& \rho \delta_{ij} \delta_{km} + \tau(\delta_{ik}\delta_{jm} + \delta_{im}\delta_{jk}) \,, \\
\mathcal{A}_{ijklmn} =& r \delta_{ij}\delta_{kl}\delta_{mn} + s\big(\delta_{ij}(\delta_{km}\delta_{ln} + \delta_{kn}\delta_{lm}) + \delta_{kl}(\delta_{im}\delta_{jn} + \delta_{in}\delta_{jm}) + \delta_{mn}(\delta_{ik}\delta_{jl} + \delta_{il}\delta_{jk})\big) \\
& + t(\delta_{ik}\delta_{jm}\delta_{ln} + \delta_{jk}\delta_{im}\delta_{ln} + \delta_{il}\delta_{jm}\delta_{kn} + \delta_{jl}\delta_{im}\delta_{kn} \\
& \quad + \delta_{ik}\delta_{jn}\delta_{lm} + \delta_{jk}\delta_{in}\delta_{lm} + \delta_{il}\delta_{jn}\delta_{km} + \delta_{jl}\delta_{in}\delta_{km}) \,.
\end{aligned}
\tag{B.13}
$$

Tracelessness requirements imply

$$
\begin{aligned}
\alpha + \beta(d-1) &= 0 \,, \\
\beta + (d-1)\delta + 2\epsilon &= 0 \,, \\
\gamma + (d-1)\rho + 2\tau &= 0 \,, \\
(d-1)r + \delta + 4s &= 0 \,, \\
(d-1)s + 4t + \epsilon &= 0 \,.
\end{aligned}
\tag{B.14}
$$

On the other hand, from the conservation of the stress-energy tensor we have

$$
\begin{aligned}
dr + (d+4)s + 2t + 2\tau &= 0 \,, \\
-dr + (d-2)s + 2(d+4)t + 2\rho &= 0 \,.
\end{aligned}
\tag{B.15}
$$

These relations leave three independent coefficients for the conserved and traceless symmetric three-point function, as expected from the analysis in the general frame. Equations relating the expressions in the general and collinear frames can be found in [28].

### B.3 Constraints of the three-point function

This subsection includes details on the constraints on these three parameters for free and interacting CFTs discussed in Section 3.3.

Table 2: Independent parameters of the three-point function of the stress-energy tensor for free boson, fermion, and tensor field theories in the general frame.

|  | Bosons | Fermions | Tensors |
|---|---|---|---|
| $a$ | $n_s \frac{1}{8} \frac{d^3}{(d-1)^3} \frac{1}{\Omega_{d-1}^3}$ | $0$ | $-\frac{d^3}{4(d-3)} \frac{n_t}{\Omega_{d-1}^3}$ |
| $b$ | $-n_s \frac{1}{8} \frac{d^4}{(d-1)^3} \frac{1}{\Omega_{d-1}^3}$ | $-\frac{2^{d/2}d^2 n_f}{16\Omega_{d-1}^3}$ | $-\frac{(d-4)d^3}{4(d-3)} \frac{n_t}{\Omega_{d-1}^3}$ |
| $c$ | $-n_s \frac{1}{8} \frac{d^2(d-2)^2}{(d-1)^3} \frac{1}{\Omega_{d-1}^3}$ | $-\frac{2^{d/2}d^2 n_f}{8\Omega_{d-1}^3}$ | $\frac{(d-2)d^3}{2(d-3)} \frac{n_t}{\Omega_{d-1}^3}$ |

Table 3: Independent parameters of the three-point function of the stress-energy tensor for free boson, fermion, and tensor field theories in the collinear frame.

|  | Bosons | Fermions | Tensors |
|---|---|---|---|
| $r$ | $n_s \frac{1}{8} \frac{d^3+28d-16}{(d-1)^3} \frac{1}{\Omega_{d-1}^3}$ | $-\frac{2^{d/2} n_f}{2\Omega_{d-1}^3}$ | $-\frac{3d^2}{(d-3)} \frac{n_t}{\Omega_{d-1}^3}$ |
| $s$ | $n_s \frac{1}{8} \frac{d(d^2-8d+4)}{(d-1)^3} \frac{1}{\Omega_{d-1}^3}$ | $\frac{d2^{d/2} n_f}{8\Omega_{d-1}^3}$ | $\frac{d^3}{2(d-3)} \frac{n_t}{\Omega_{d-1}^3}$ |
| $t$ | $n_s \frac{1}{8} \frac{d^3}{(d-1)^3} \frac{1}{\Omega_{d-1}^3}$ | $0$ | $-\frac{d^3}{4(d-3)} \frac{n_t}{\Omega_{d-1}^3}$ |

### B.3.1 Free theories

Tables 2 and 3 show the three independent parameters of the three-point functions of the stress-energy tensor in free boson, fermion, and tensor theories in the general and collinear frames [28,38]. In these tables, $n_s$ is the number of scalar fields, $n_f$ is the number of fermionic fields and $n_t$ is the number of degrees of freedom contributed by the tensor fields (which we will only consider in even dimensions).

For free theories, $C_T$ is [28,38]

$$C_T^s = n_s \frac{d}{d-1} \frac{1}{\Omega_{d-1}^2}, \qquad C_T^f = n_f \frac{d}{2} 2^{d/2} \frac{1}{\Omega_{d-1}^2}, \qquad C_T^t = n_t \frac{d^2}{\Omega_{d-1}^2}. \qquad (B.16)$$

### B.3.2 Average null energy condition in CFTs

The *averaged null energy condition* (ANEC) states that the integral of the expectation value of the null energy along a complete null worldline is non-negative

$$\int_{-\infty}^{\infty} dx^- \langle T_{--}(x^-) \rangle_\psi \geq 0. \qquad (B.17)$$

For CFTs, ANEC was derived for a special class of states in [35]. Refs. [11,53] provided a more general proof of ANEC using causality arguments. Also, using causality and bootstrap methods, [39] proved ANEC for any unitary parity-preserving CFT with a unique energy-momentum tensor.

By assuming ANEC, Hofman and Maldacena [40] derived in $d = 4$ some constraints on the anomaly coefficients, known as *conformal collider bounds*, which lead to bounds on the parameters that characterize any $\langle TTT \rangle$. Ref. [38] extended these results to general $d$. They considered the integrated energy flux measured per unit angle in the transverse directions at a large sphere of radius $r$

$$\mathcal{E}(\vec{n}) = \lim_{r \to \infty} r^2 \int_{-\infty}^{\infty} dx^0 n^i T_i^0(t, r\vec{n}^i), \qquad (B.18)$$

where $n^i$ is a unit vector in $\mathbb{R}^{d-1}$ and specifies the point on the $S^{d-2}$ at infinity where we are measuring. For states defined by the action of a scalar operator in the vacuum $\mathcal{O}|0\rangle$, the ANEC tells us

$$\langle \mathcal{E}(\vec{n}) \rangle = \frac{\langle 0 | \mathcal{O}^{\dagger} \mathcal{E}(\vec{n}) \mathcal{O} | 0 \rangle}{\langle 0 | \mathcal{O}^{\dagger} \mathcal{O} | 0 \rangle} \geq 0. \tag{B.19}$$

They considered the states $\mathcal{O} \sim \epsilon^{\mu\nu} T_{\mu\nu}$, where $\epsilon_{\mu\nu}$ is the polarization tensor. Because $T^{\mu\nu}$ is conserved we can choose $\epsilon$ to point in spacelike directions $\epsilon_{ij}$ [40]. The most general form of the energy flux at null infinity in the direction indicated by $\vec{n}$ is [38, 40]

$$\begin{aligned} \langle \mathcal{E}(\vec{n}) \rangle &= \frac{\langle 0 | \epsilon_{ij}^* T^{ij} \mathcal{E}(\theta) \epsilon_{lk}^* T^{lk} | 0 \rangle}{\langle 0 | \epsilon_{ij}^* T^{ij} \epsilon_{lk}^* T^{lk} | 0 \rangle} \\ &= \frac{E}{\Omega_{d-2}} \left[ 1 + t_2 \left( \frac{\epsilon_{ij}^* \epsilon_{il} n^j n^l}{\epsilon_{ij}^* \epsilon_{ij}} - \frac{1}{d-1} \right) + t_4 \left( \frac{|\epsilon_{ij} n^i n_j|^2}{\epsilon_{ij}^* \epsilon_{ij}} - \frac{2}{d^2-1} \right) \right], \end{aligned} \tag{B.20}$$

where $E$ is the total energy. The structure of this expression is the result of conformal symmetries and has two undetermined coefficients in agreement with the results in Appendix B.2 (we found three free coefficients, one determined by the Ward identity). The negative terms appearing in the two factors multiplied by $t_2$ and $t_4$ lead to constraints on the $t_2$ and $t_4$. We can write these coefficients in terms of $a$, $b$ and $c$ [38]

$$\begin{aligned} t_2 &= \frac{2(d+1)}{d} \frac{a(d-1)(d(d+8)+4) + 3bd^2 - cd(2d+1)}{a(d(d+1)-6) - c(d+1) - 2b}, \\ t_4 &= \frac{(d+1)(d+2)}{d} \frac{3a(d(1-2d)+1) - 2bd^2 + cd(d+1)}{a(d(d+1)-6) - c(d+1) - 2b}. \end{aligned} \tag{B.21}$$

### B.3.3 Constraints on $\langle TTT \rangle$

Demanding $\langle \mathcal{E} \rangle$ to be non-negative imposes some constraints on the coefficients $t_2$ and $t_4$ [38, 40]

$$\begin{aligned} 1 - \frac{1}{d-1} t_2 - \frac{2}{d^2-1} t_4 &\geq 0, \\ \left( 1 - \frac{1}{d-1} t_2 - \frac{2}{d^2-1} t_4 \right) + \frac{1}{2} t_2 &\geq 0, \\ \left( 1 - \frac{1}{d-1} t_2 - \frac{2}{d^2-1} t_4 \right) + \frac{d-2}{d-1} (t_2 + t_4) &\geq 0. \end{aligned} \tag{B.22}$$

Using (B.21), these constraints can be written in terms of $a$, $b$ and $c$

$$\begin{aligned} a(d(d+4)-4) + d(2b-c) &\leq 0, \\ a(d(d+6)-8) + b(3d-2) - 2cd &\geq 0, \\ 4a + 2b - c &\leq 0. \end{aligned} \tag{B.23}$$

In addition, the positivity of the central charge $C_T$ leads to

$$(d-2)(d+3)a - 2b - (d+1)c > 0. \tag{B.24}$$

For collinear points, (B.23) imply

$$\begin{aligned} \frac{8\gamma (d(d^2+d-10)+4)\Omega_{d-1} - 8\alpha d \Omega_{d-1} - C_T d(d((d-1)d(d+5)-28)+20)}{(d-3)(d-2)} &\leq 0, \\ \frac{2(\alpha d + \gamma(d(d+7)-4))\Omega_{d-1} - C_T d(d(d+5)-2)}{d-2} &\geq 0, \\ \frac{d(\alpha+4\gamma)\Omega_{d-1} - 4\gamma \Omega_{d-1} - C_T(d-1)d}{(d-2)(d-1)} &\leq 0. \end{aligned} \tag{B.25}$$

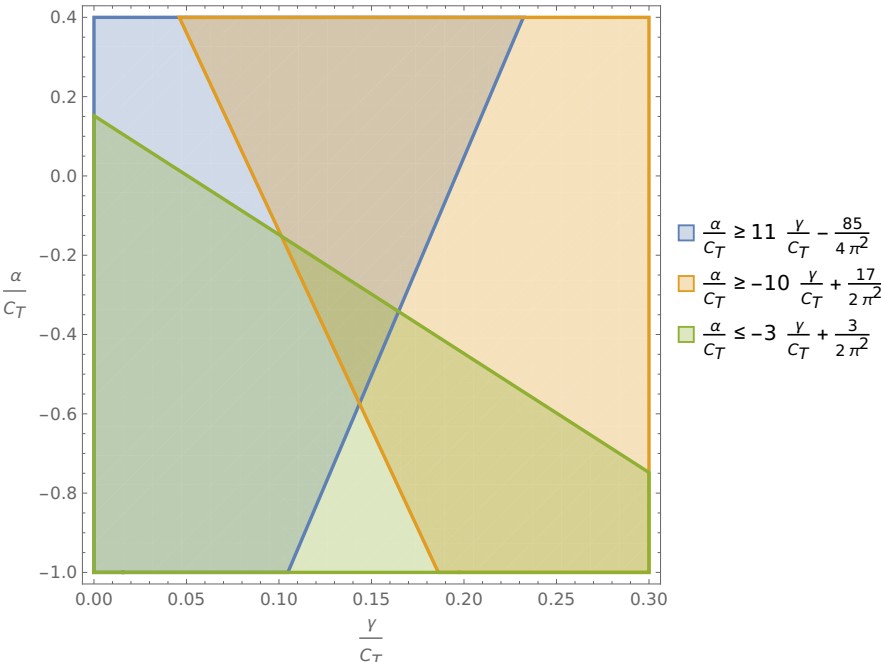

Figure 4: Possible values of $\alpha/C_T$ and $\gamma/C_T$ according to constraints (B.25).

Note that for $d = 2$ (B.25) are not defined and for $d = 3$ the first condition is not defined. The constraints on $\alpha/C_T$ and $\gamma/C_T$ for $d = 4$ are plotted in Fig. 4.

As a check, we apply the constraints to the free scalar theory, where have

$$\alpha = -\frac{(d^5 + 4d^4 - 17d^3 + 36d - 16)n_s}{8(d-1)^2\Omega_{d-1}^3}, \quad \gamma = \frac{(d^4 + d^3 - 4d^2 + 4d)n_s}{8(d-1)^2\Omega_{d-1}^3}, \quad C_T = \frac{n_s}{\Omega_{d-1}^2}\frac{d}{d-1}. \quad \text{(B.26)}$$

Substitution of these expressions in (B.25) shows that the three constraints are verified.

We go back to the general frame and consider an interacting CFT whose $\langle TTT \rangle$ is characterized by $n_s, n_f, n_t$ instead of $a, b, c$. In terms of these parameters, the constraints (B.23) are

$$-\frac{(d^2 - 4)d^3}{d - 3}n_t \leq 0,$$

$$\frac{1}{2}(d + 2)d^2 n_f \geq 0, \quad \text{(B.27)}$$

$$-\frac{(d^2 - 4)d^2}{2(d - 1)^3}n_s \leq 0.$$

For $d \geq 4$, constraints (B.27) imply that $(n_s, n_f, n_t)$ are non-negative.

## C  Connected correlators

### C.1  Combinatorics

Consider the two-point function $\langle (\mathcal{O}_h)^{p\dagger}(\mathcal{O}_h)^p \rangle$ of the multi-trace operator defined in (81) and say we wish to factor this into all possible connected (2a)-point functions of some maximum order: $1 \leq a \leq a_{\max}$. We can organize this as follows. Let $\{m_a\}_{a=1,\dots,a_{\max}}$ be a collection of

natural numbers satisfying

$$\sum_{a=1}^{a_{max}} a\, m_a = p\,. \tag{C.1}$$

Then we write

$$\langle (\mathcal{O}_h)^{p\dagger}(\mathcal{O}_h)^p \rangle = \sum_{\{m_a\}}^{p} \langle (\mathcal{O}_h)^{p\dagger}(\mathcal{O}_h)^p \rangle \Big|_{\{m_a\}} + \dots\,, \tag{C.2}$$

where $\sum_{\{m_a\}}^{p}$ denotes summing over all sets $\{m_a\}$ satisfying (C.1) and $\langle (\mathcal{O}_h)^{p\dagger}(\mathcal{O}_h)^p \rangle \Big|_{\{m_a\}}$ is the term in the $2p$-point function consisting of $m_1$ two-point functions, $m_2$ four-point functions, and so on. The $\dots$ indicates connected correlators of order greater than $(2a_{max})$.

Let us first focus on a particular $\langle (\mathcal{O}_h)^{p\dagger}(\mathcal{O}_h)^p \rangle \Big|_{\{m_a\}}$ and ask how many ways it can break up into its connected part. First will pull out all the terms that need to go into 2-point functions, then from the remainder the terms that go into 4-point functions, then 6-point functions, and so on. The ways of doing so are

$$
\begin{aligned}
\langle (\mathcal{O}_h)^{p\dagger}(\mathcal{O}_h)^p \rangle \Big|_{\{m_a\}} &= \binom{p}{m_1}^2 \langle (\mathcal{O}_h)^{m_1\dagger}(\mathcal{O}_h)^{m_1} \rangle_{(2)} \binom{p-m_1}{2m_2}^2 \langle (\mathcal{O}_h)^{2m_2\dagger}(\mathcal{O}_h)^{2m_2} \rangle_{(4)} \cdots \\
&= \prod_{a=1}^{a_{max}} \binom{p - \sum_{b=0}^{a-1} b m_b}{a m_a}^2 \langle (\mathcal{O}_h)^{a m_a\dagger}(\mathcal{O}_h)^{a m_a} \rangle_{(2a)} \\
&= (p!)^2 \prod_{a=1}^{a_{max}} \frac{\langle (\mathcal{O}_h)^{a m_a\dagger}(\mathcal{O}_h)^{a m_a} \rangle_{(2a)}}{(a m_a)!^2}\,,
\end{aligned}
\tag{C.3}
$$

where the subscript on the $\langle (\mathcal{O}_h)^{a m_a\dagger}(\mathcal{O}_h)^{a m_a} \rangle_{(2a)}$ indicates that we still need to break up this higher-point correlator entirely into connected $(2a)$-point functions. Let us do so now.

We build $(2a)$-point functions from $\langle (\mathcal{O}_h)^{a m_a\dagger}(\mathcal{O}_h)^{a m_a} \rangle_{(2a)}$ sequentially. The left-most $\mathcal{O}_h^\dagger$ has a choice of $(a m_a - 1$ choose $a - 1)$ $\mathcal{O}_h^\dagger$'s and $(a m_a$ choose $a)$ $\mathcal{O}_h$'s to form a $\langle (\mathcal{O}_h)^{a\dagger}(\mathcal{O}_h)^a \rangle$ leaving behind a $\langle (\mathcal{O}_h)^{a(m_a-1)\dagger}(\mathcal{O}_h)^{a(m_a-1)} \rangle_{(2a)}$. Proceeding along we find

$$
\langle (\mathcal{O}_h)^{a m_a\dagger}(\mathcal{O}_h)^{a m_a} \rangle_{(2a)} = \left[ \prod_{n=0}^{m_a-1} \binom{a(m_a-n)-1}{a-1} \binom{a(m_a-n)}{a} \right] \times \langle (\mathcal{O}_h)^{a\dagger}(\mathcal{O}_h)^a \rangle_{conn}^{m_a}\,. \tag{C.4}
$$

We can simplify by noting that the product of the second terms in the bracket is

$$
\prod_{n=0}^{m_a-1} \binom{a(m_a-n)}{a} = \frac{(a m_a)!}{(a!)^{m_a}}\,, \tag{C.5}
$$

while the product of the first term in the brackets is

$$
\begin{aligned}
\prod_{n=0}^{m_a-1} \binom{a(m_a-n)-1}{a-1} &= \frac{(a m_a)(a m_a-1)\dots(a m_a-a+1)\times(a m_a-a)\times(a m_a-a-1)\dots}{((a-1)!)^{m_a}\times(a m_a)(a m_a-a)\dots} \\
&= \frac{(a m_a)!}{(a!)^{m_a} m_a!}\,.
\end{aligned}
\tag{C.6}
$$

Putting this in (C.3) we find the relatively simple expression

$$
\langle (\mathcal{O}_h)^{p\dagger}(\mathcal{O}_h)^p \rangle \Big|_{\{m_a\}} = (p!)^2 \times \left[ \prod_{a=1}^{a_{max}} \frac{\langle (\mathcal{O}_h)^{a\dagger}(\mathcal{O}_h)^a \rangle_{conn}^{m_a}}{(a!)^{2 m_a} m_a!} \right] = (p!)^2 \langle \mathcal{O}_h^\dagger \mathcal{O}_h \rangle^p \times \left[ \prod_{a=1}^{a_{max}} \frac{x_a^{m_a}}{m_a!} \right],\tag{C.7}
$$

with

$$x_a = \frac{\langle (\mathcal{O}_h)^{a\dagger}(\mathcal{O}_h)^a \rangle_{\text{conn}}}{(a!)^2 \langle \mathcal{O}_h^\dagger \mathcal{O}_h \rangle^a} \, . \tag{C.8}$$

Note that $x_1 = 1$ while generically $x_a \sim N^{2-2a}$ which is independent of how we normalize the $\langle \mathcal{O}\mathcal{O} \rangle$ two-point function.

To understand this expression we look at the following example. Let's assume we have the 8-point function $\langle (\mathcal{O}_h)^{4\dagger}(\mathcal{O}_h)^4 \rangle$. It can be decomposed in the following collections of even point functions: $\{m_1 = 1, m_2 = 0, m_3 = 1\}$, $\{m_1 = 4, m_2 = m_3 = 0\}$, $\{m_1 = 2, m_2 = 1, m_3 = 0\}$ and $\{m_1 = 0, m_2 = 2, m_3 = 0\}$. Then for one possible decomposition and using Eq. (C.7) we have

$$\langle (\mathcal{O}_h)^{4\dagger}(\mathcal{O}_h)^4 \rangle \Big|_{\{m_1=2,m_2=1,m_3=0\}} = (4!)^2 \left( \frac{\langle (\mathcal{O}_h)^\dagger (\mathcal{O}_h) \rangle^2}{2!} \right) \left( \frac{\langle (\mathcal{O}_h)^{2\dagger}(\mathcal{O}_h)^2 \rangle}{(2!)^2} \right) \tag{C.9}$$
$$= 72 \langle (\mathcal{O}_h)^\dagger (\mathcal{O}_h) \rangle^2 \langle (\mathcal{O}_h)^{2\dagger}(\mathcal{O}_h)^2 \rangle \, .$$

This result matches with the direct consideration of possible 4-point functions $\binom{4}{2}^2 = 36$ and possible two-point functions $\binom{2}{1} = 2$.

The full correlators are then given by

$$\langle (\mathcal{O}_h)^{p\dagger}(\mathcal{O}_h)^p \rangle = (p!)^2 \langle \mathcal{O}_h^\dagger \mathcal{O}_h \rangle^p \sum_{\{m_a\}} \left[ \prod_{a=1}^{a_{\max}} \frac{x_a^{m_a}}{m_a!} \right] + \dots, \tag{C.10}$$

where again the sum is over all sets $\{m_a\}$ satisfying (C.1). While this expression is compact, since the $m_a$'s are constrained (and since $x_1 = 1$) we can write $m_1 = p - \sum_{a=2}^{a_{\max}} a m_a$ to write this only in terms of the corrections that four and higher-point correlators contribute to the $(2p)$-point function:

$$\langle (\mathcal{O}_h)^{p\dagger}(\mathcal{O}_h)^p \rangle = p! \langle \mathcal{O}_h^\dagger \mathcal{O}_h \rangle^p \sum_{\{m_a\}_{a=2,\dots,a_{\max}}}^{\leq p} \frac{p!}{(p - \sum_{a=2}^{a_{\max}} a m_a)!} \left[ \prod_{a=2}^{a_{\max}} \frac{x_a^{m_a}}{m_a!} \right] + \dots, \tag{C.11}$$

where now the sum $\sum_{\{m_a\}_{a=2,\dots,a_{\max}}}^{\leq p}$ is over all possible $\{m_a\}_{a=2,\dots,a_{\max}}$ satisfying $\sum_{a=2}^{a_{\max}} m_a \leq p$.

## C.2 Asymptotics of R

A quantity of interest in Section 5 is the ratio

$$R_{a_{\max}}(\{x_a\}, p) = \frac{\sum_{\{m_a\}_{a=2,\dots,a_{\max}}}^{\leq p-1} \frac{(p-1)!}{(p - \sum_{a=2}^{a_{\max}} a m_a - 1)!} \left[ \prod_{a=2}^{a_{\max}} \frac{x_a^{m_a}}{m_a!} \right]}{\sum_{\{m_a\}_{a=2,\dots,a_{\max}}}^{\leq p} \frac{p!}{(p - \sum_{a=2}^{a_{\max}} a m_a)!} \left[ \prod_{a=2}^{a_{\max}} \frac{x_a^{m_a}}{m_a!} \right]}, \tag{C.12}$$

which appears in the ratio of $\langle (\mathcal{O}_h)^{p-1\dagger}(\mathcal{O}_h)^{p-1} \rangle$ and $\langle (\mathcal{O}_h)^{p\dagger}(\mathcal{O}_h)^p \rangle$ after breaking up into connected $(2a)$-point correlators up to a maximum order $a_{\max}$. We are interested in the large $p$, small $|x|$ asymptotics of this object as $p \sim N^\alpha$ and $x_a \sim N^{2-2a}$ in the large $N$ limit. As discussed in Section 5, we will keep $\alpha < \frac{2a_{\max}}{a_{\max}+1} < 2$.

Let us call a term of fixed $\{m_a\}$ in the numerator or denominator $n_{\{m_a\}}$ and $d_{\{m_a\}}$, respectively so that

$$R_{a_{\max}}(\{x_a\}, p) = \frac{\sum_{\{m_a\}}^{\leq p-1} n_{\{m_a\}}}{\sum_{\{m_a\}}^{\leq p} d_{\{m_a\}}} \, . \tag{C.13}$$

We note that for almost all configurations of $\{m_a\}$

$$n_{\{m_a\}} = \left(1 - \frac{\sum_{a=2}^{a_{max}} a\, m_a}{p}\right) d_{\{m_a\}}\,. \tag{C.14}$$

Of course, terms where $\sum_a a\, m_a = p$ are explicitly excluded from the numerator sum, however we can still write (C.14) as such terms would be exactly zero anyways. We then find

$$R_{a_{max}}(\{x_a\}, p) = 1 - \frac{\sum_{\{m_a\}}^{\leq p} \left(\frac{\sum_a a\, m_a}{p}\right) d_{\{m_a\}}}{\sum_{\{m_a\}}^{\leq p} d_{\{m_a\}}}\,. \tag{C.15}$$

It is this second term that is now our focus and we wish to show that it is suppressed by a negative power of $N$ as $p \sim N^{\alpha}$. To see that this is, we note that the terms $d_{\{m_a\}}$ will have different $N$-scalings depending on the combinatorics determined by $\{m_a\}$. There will be some term(s) of maximal $N$-scaling that will dominate this sum. However, if those terms occur for any choice of $\{m_a\}$ such that $\sum_a a\, m_a \nsim p$ then the numerator is parametrically suppressed and we are done. The only case we need to check is when $\sum_a a\, m_a \sim p$ in $N$-scaling and verify that the corresponding $d_{\{m_a\}}$ is not maximal.

To that end, let there be a collection $\{m_a\}$ such that $\sum_a a\, m_a = p - r$ where the remainder, $r \sim N^{\beta}$, has subleading scaling compared to $p$: $\beta < \alpha$. If $a_{max} \sim O(1)$, then there must be at least one (although possibly multiple) term with $a\, m_a \sim p$. For simplicity we will assume that one term, $\bar{m}_{\bar{a}}$, dominates and satisfies $\bar{a}\bar{m}_{\bar{a}} = p - r$. The case with multiple dominant terms follows similarly. Implementing Stirling's approximation, the leading scaling to $d_{\{m_a\}}$ is

$$d_{\{m_a\}} \sim \sqrt{\frac{p}{\bar{m}_{\bar{a}}\, r}} \frac{p^p}{\bar{m}_{\bar{a}}^{\bar{m}_{\bar{a}}} r^r} N^{\sum_a (2-2a)m_a} \sim N^{\alpha p - \alpha \bar{m}_{\bar{a}} - \beta r + \sum_a (2-2a)m_a} \\ \sim N^{(\alpha-2)(\bar{a}-1)\bar{m}_{\bar{a}}}\,, \tag{C.16}$$

where in the second line we have dropped all terms in the exponent that are themselves not leading in $N$. Since $\bar{a} \geq 2$ and $\alpha < 2$, this term is doubly-exponentially suppressed and cannot be maximal.

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
