# Peer review of "How negative can null energy be in large N CFTs?"

_SciPost Physics, doi:SciPost Phys. 19, 087 (2025)_

## Round 1 · Referee Report · Anonymous (Referee 2) · 2025-6-6

Strengths

  1. Clearly states the problem addressed and methods used, and appears technically sound
  2. Makes use of a variety of CFT models and techniques to provide evidence of conjecture

Weaknesses

  1. The physical motivation for studying the specific energy inequalities that are being studied is not very clear. The paper does not give any examples of physical consequences that would follow from their "strong" conjecture.
  2. The work is looking at a special class of states in a special class of theories in order to support a conjecture. The space of quantum field theories and states within them is vast, so this is largely unavoidable, but it also means that the danger of getting fooled by "looking under the lamppost" is high.

Report

My main comment is about the motivation for the question that is addressed in the paper. I sympathize with the motivation from general theoretical curiosity, but are there any physical questions that hinge on the "strong" conjecture posted by the authors? For example, the averaged null energy condition (ANEC) became a hot topic after Hoffman and Maldacena pointed out that it could be used to prove interesting general statements about conformal field theories in general spacetime dimensions. I understand that this is an exploratory work, but I believe that a serious attempt should be made to grapple with this and convey the motivation to the reader. To be frank, this paper reads like it is addressed to an in-group that has already agreed that any result about energy inequalities in quantum field theory is interesting.

The general motivation that is conventionally given for studying energy inequalities in quantum field theory (and indeed it is mentioned in this paper) comes from the fact that classical general relativity requires various energy positivity constraints to avoid pathological solutions. However, the path from quantum energy inequalities to any kind of statement about macroscopic general relativity is not at all clear, even in broad outline. The case of a non-minimally coupled scalar is a good example of this. As the authors know well from their own previous work (Ref. 17) a 4D scalar field with a non-minimal coupling to gravity violates energy conditions even classically. However, it does not give rise to any pathological solution because we can simply go to Einstein frame, where the non-minimal coupling to gravity is replaced by a scalar potential. While this is certainly a special example (a theory with a scalar operator of dimension 2, allowing a marginal coupling to gravity), it illustrates the point that the consistency of a theory coupled to gravity will necessarily involve the mixing of the gravitational and non-gravitational degrees of freedom, and that this need not be dominated by short-distance quantum effects. In fact, back-reaction is generically important in pathological solutions of the kind that often result when energy inequalities are violated. (For example, it is often the stability rather than the existence of these solutions that is problematic.)

I get that these are difficult questions to answer, and I am not asking the
authors to provide a general solution in this paper. However, as someone not working in this subject, I would like to see a little more discussion of these points.

A somewhat related comment is that the first page and a half of the introduction focuses on the authors' own work, rather than giving a broad overview of the field. I was surprised that there was no mention of the ANEC, since it is both a limiting case of the energy condition studied by the authors, and a perfect example of a "success story" for this kind of work. Namely, the ANEC was shown to be related to interesting constraints on CFTs, and this led to further theoretical work proving it in general CFTs.

The rest of the paper is well written and appears to be technically sound. I would be more cautious about saying that the results in the paper "suggest" various more general results, but the statements in the paper are clear and the reader can make up their own mind. I believe that overall this paper is above threshold for publication.

Requested changes

  1. Make a serious attempt to address the motivation issues in the report above. I suggest that the audience that the authors keep in mind is a wider audience of theorists interested in structural questions in quantum field theory.

Recommendation

Ask for minor revision

  • validity: high
  • significance: ok
  • originality: good
  • clarity: high
  • formatting: perfect
  • grammar: perfect

Author:  Diego Pardo Santos  on 2025-07-03  [id 5615]

(in reply to Report 2 on 2025-06-06)

We thank referee 2, for their careful reading, their insightful questions, and their suggestions for improvement of the manuscript. In addressing referee 2’s concerns we have made moderate revisions to our article to establish better the context of our results.

“My main comment is about the motivation for the question that is addressed in the paper. I sympathize with the motivation from general theoretical curiosity, but are there any physical questions that hinge on the "strong" conjecture posted by the authors?”

The primary motivation for studying the conditions averaged over spacetime regions of the form of this paper (what are referred to as QEIs) is their use in constraining semi-classical gravity coupled to quantum fields. In particular, our work builds upon prior works establishing that QEIs in both state-dependent and state-independent form are useful data in constraining solutions to the Einstein equation, e.g. long traversable wormholes. We recognize that much of this context and motivation was missing or unclear and we have made substantial additions to our introduction and our discussion sections to both bring this background to reader’s attention and to cast the current work in that context. We thank the referee for bringing this to our attention.

“The case of a non-minimally coupled scalar is a good example of this. As the authors know well from their own previous work (Ref. 17) a 4D scalar field with a non-minimal coupling to gravity violates energy conditions even classically. However, it does not give rise to any pathological solution because we can simply go to Einstein frame, where the non-minimal coupling to gravity is replaced by a scalar potential… it illustrates the point that the consistency of a theory coupled to gravity will necessarily involve the mixing of the gravitational and non-gravitational degrees of freedom…”

The referee is correct to point out that the NMC scalar is an example that illustrates well the subtleties in how energy conditions are applied in semi-classical gravity. In addressing the question of physical relevance of the Jordan vs. the Einstein frame, we should really treat semi-classical gravity as an effective description of a theory where both gravity and the source matter are quantum as the map between the two frames mixes gravitational and scalar degrees of freedom. In that context, the scalar potential generated in the Einstein frame contains a tower of irrelevant couplings, which, when large (such as in states with large values of the scalar field), signal a breakdown of effective field theory. This is further complicated by the fact (and primary motivation of the current work) that the status of QEIs for interacting theories is not well understood. Our previous work mentioned by the referee also establishes similar breakdowns of gravitational effective field theory in the Jordan frame for large scalar field values. These comments serve as physical context and motivation for the state-dependent bounds (such as that obeyed by the NMC scalar): the operator expectation values upon which the bound depends act as parameters demarcating the effective field theory in which negative energy densities are bounded and semi-classical gravity is well behaved. We have added a paragraph under equation (6) to this effect to establish context for the state-dependent bounds.

“A somewhat related comment is that the first page and a half of the introduction focuses on the authors' own work, rather than giving a broad overview of the field. I was surprised that there was no mention of the ANEC, since it is both a limiting case of the energy condition studied by the authors, and a perfect example of a "success story" for this kind of work. Namely, the ANEC was shown to be related to interesting constraints on CFTs, and this led to further theoretical work proving it in general CFTs.”

We thank the referee for pointing out this omission. We agree with the referee that it is an important result for establishing the context of energy conditions in quantum theories and to also motivate the QEIs of study in our manuscript, and highlight their differences with the ANEC. We have added a discussion of the ANEC early in the introduction, starting in the second paragraph and continuing onward through the third paragraph. In the course of this discussion we have referenced more prior works on energy conditions and geometric theorems which we hope establishes a broader context for our work.

Changes to the most recent version of our manuscript are highlighted in blue and those addressing referee 2’s comments are listed as follows:

  1. Expanded the introduction, starting at the middle of the second paragraph to establish the context of singularity theorems, ANEC, and QEIs.
  2. Added paragraph under equation (6) on the role of state-dependent bounds in effective field theory.
  3. Added sentence on ANEC in paragraph starting in line 131.
  4. Added sentence in paragraph starting on line 188 about the consequences of the strong conjecture.
  5. Added a new portion in the discussion section (starting line 864) establishing our results in the context of singularity theorems and wormhole spacetimes.

---

## Round 1 · Referee Report · Anonymous (Referee 1) · 2025-6-6

Report

This paper investigates the possibility of having states with negative smeared null energy, particularly in the large N limit of CFTs. This is an interesting question, since little is understood about energy conditions in QFT in general, especially interacting theories in higher dimensions. This paper is written well, but I have some suggestions that would make it clearer and some clarifications I would like to be made before it can be published.

1) The authors make use of a smearing function g(x) in their calculations. They mention some conditions (smooth, real, compact support) that were used for the 2d condition in equation (2) of the introduction, and it later sections it seems as though the same conditions are used in the higher dimension calculations, but for clarity it would be useful to have the general conditions on g(x) in higher dimensions stated explicitly in the introduction.

2) In line (159) there is a grammatical error, I suspect "diverges" should be "divergences".

3) In line 290, for clarity this should be changed to: "there is one operator guaranteed to exist in every \emph{local} CFT: the stress-energy tensor..."

4) There are two different definitions of the state $|\psi\rangle$, one in equation (44) and one in equation (51). The notation for one of these should be changed.

5) In line 477 the state $|h^p_∆\rangle$ is referenced, but upon initial reading it was not clear to me that this was the first introduction of this definition of the state. I think changing the language to be something like “Alternatively we could define another state $|h^p_∆\rangle$...” or “Alternatively we could define a different state $|h^p_∆\rangle$...” would make this clearer and distinguish it from $|\mathcal{O}_h^p\rangle$.

6) Can the authors comment on the contributions of contact terms in general for their analysis? Are there assumptions on the smearing functions that ensure that contact terms do not appear? And is it clear they would not appear after analytic continuation (as they do in e.g. 2309.14409)?

Requested changes

I recommend the paper for publication after the above points have been addressed.

Recommendation

Ask for minor revision

  • validity: -
  • significance: -
  • originality: -
  • clarity: -
  • formatting: -
  • grammar: -

Author:  Diego Pardo Santos  on 2025-07-03  [id 5614]

(in reply to Report 1 on 2025-06-06)

We thank the referee for their careful reading, their insightful questions, and their suggestions for improvement of the manuscript. In light of their suggestions we have made minor revisions to our article to clear up notational confusions and to address their questions.

While the 2d QEI explicitly states that the smearing function is of compact support this is actually more than necessary. For the purposes of our results it is sufficient that the function falls off sufficiently fast (for instance, a Gaussian serves a perfectly suitable smearing function). Note that if the smearing function is not of that kind the bound diverges but the inequality is still true. We have added a footnote on page 2 stating what is meant by “smearing function.” The typo has been corrected. We thank the referee for this clarification and it has been implemented. To avoid equivocation, we have renamed the state in equation (51) to $|\Phi\rangle$. Upon further analysis, it is not clear that the state $|h_\Delta^p\rangle$ defined in that paragraph yields the same correlation functions as $|\mathcal O_\Delta^p\rangle$ due to possible Wick contractions of operators preparing the state. As this state does not play any role in our analysis and to prevent distraction from the discussion of this section we have decided to remove the mention of $|h_\Delta^p\rangle$. We thank the referee for this insightful question. The absence of contact terms in operator OPEs is an assumption in our analysis. At the level of states prepared by scalar operators in Section 2 this assumption is well justified by being in an interacting large-N theory: by dimensional analysis, contact terms can only appear when the scalar operator has integer dimension which then implies that it is a composite of a free field. For the stress tensor states of Section 3, this is genuinely an assumption. The relaxation of this assumption however, does not affect our large-N analysis: by construction, our states always result in two- and three-point correlators where the stress tensors appear at separate points.

A list of changes to the article (highlighted in blue) addressing referee 1’s comments is as follows:

  1. Added footnote on page 3 clarifying what is meant by “smearing function.”
  2. “diverges” to “divergences” in line 196.
  3. Added “local” to line 327.
  4. Renamed state $|\psi\rangle$ to $|\Phi\rangle$ in equations (51) and (52).
  5. Removed mention of the state $|h_\Delta^p\rangle$ from paragraph below equation (81).
  6. Added sentence in line 344 on the assumption of no contact terms.

---

## Round 2 · Referee Report · Anonymous (Referee 1) · 2025-7-21

Report

I thank the authors for their updated version, which I think is much clearer. With these updates, I recommend the paper for publication.

Recommendation

Publish (meets expectations and criteria for this Journal)

---

## Round 2 · Referee Report · Anonymous (Referee 2) · 2025-8-5

Report

I thank the authors for the updates. I believe they have addressed the points I made in my first report.

Recommendation

Publish (meets expectations and criteria for this Journal)

---

## Round 2 · Author Response

Dear Editor-in-charge.

We thank the referees, for their careful reading, their insightful questions, and their suggestions for improvement of the manuscript.

Reply to referee 1:

In light of the suggestions of referee 1, we have made minor revisions to our article to clear up notational confusions and to address their questions.

1. While the 2d QEI explicitly states that the smearing function is of compact support this is actually more than necessary. For the purposes of our results it is sufficient that the function falls off sufficiently fast (for instance, a Gaussian serves a perfectly suitable smearing function). Note that if the smearing function is not of that kind the bound diverges but the inequality is still true. We have added a footnote on page 2 stating what is meant by “smearing function.”
2. The typo has been corrected.
3. We thank the referee for this clarification and it has been implemented.
4. To avoid equivocation we have renamed the state in equation (51) to $|\Phi\rangle$.
5. Upon further analysis, it is not clear that the state $|h_\Delta^p\rangle$ defined in that paragraph yields the same correlation functions as $|\mathcal O_\Delta^p\rangle$ due to possible Wick contractions of operators preparing the state. As this state does not play any role in our analysis and to prevent distraction from the discussion of this section we have decided to remove the mention of $|h_\Delta^p\rangle$.
6. We thank the referee for this insightful question. The absence of contact terms in operator OPEs is an assumption in our analysis. At the level of states prepared by scalar operators in Section 2 this assumption is well justified by being in an interacting large-N theory: by dimensional analysis contact terms can only appear when the scalar operator has integer dimension which then implies that it is a composite of a free field. For the stress tensor states of Section 3, this is genuinely an assumption. The relaxation of this assumption however does not affect our large-N analysis: by construction our states always result in two- and three-point correlators where the stress tensors appear at separate points.

Reply to referee 2:

In addressing referee 2’s concerns we have made moderate revisions to our article to establish better the context of our results.   “My main comment is about the motivation for the question that is addressed in the paper. I sympathize with the motivation from general theoretical curiosity, but are there any physical questions that hinge on the "strong" conjecture posted by the authors?”

The primary motivation for studying the conditions averaged over spacetime regions of the form of this paper (what are referred to as QEIs) is their use in constraining semi-classical gravity coupled to quantum fields. In particular, our work builds upon prior works establishing that QEIs in both state-dependent and state-independent form are useful data in constraining solutions to the Einstein equation, e.g. long traversable wormholes. We recognize that much of this context and motivation was missing or unclear and we have made substantial additions to our introduction and our discussion sections to both bring this background to reader’s attention and to cast the current work in that context. We thank the referee for bringing this to our attention.

“The case of a non-minimally coupled scalar is a good example of this. As the authors know well from their own previous work (Ref. 17) a 4D scalar field with a non-minimal coupling to gravity violates energy conditions even classically. However, it does not give rise to any pathological solution because we can simply go to Einstein frame, where the non-minimal coupling to gravity is replaced by a scalar potential… it illustrates the point that the consistency of a theory coupled to gravity will necessarily involve the mixing of the gravitational and non-gravitational degrees of freedom…”

The referee is correct to point out that the NMC scalar is an example that illustrates well the subtleties in how energy conditions are applied in semi-classical gravity. In addressing the question of physical relevance of the Jordan vs. the Einstein frame, we should really treat semi-classical gravity as an effective description of a theory where both gravity and the source matter are quantum as the map between the two frames mixes gravitational and scalar degrees of freedom. In that context, the scalar potential generated in the Einstein frame contains a tower of irrelevant couplings, which, when large (such as in states with large values of the scalar field), signal a breakdown of effective field theory. This is further complicated by the fact (and primary motivation of the current work) that the status of QEIs for interacting theories is not well understood. Our previous work mentioned by the referee also establishes similar breakdowns of gravitational effective field theory in the Jordan frame for large scalar field values. These comments serve as physical context and motivation for the state-dependent bounds (such as that obeyed by the NMC scalar): the operator expectation values upon which the bound depends act as parameters demarcating the effective field theory in which negative energy densities are bounded and semi-classical gravity is well behaved. We have added a paragraph under equation (6) to this effect to establish context for the state-dependent bounds.

“A somewhat related comment is that the first page and a half of the introduction focuses on the authors' own work, rather than giving a broad overview of the field. I was surprised that there was no mention of the ANEC, since it is both a limiting case of the energy condition studied by the authors, and a perfect example of a "success story" for this kind of work. Namely, the ANEC was shown to be related to interesting constraints on CFTs, and this led to further theoretical work proving it in general CFTs.”

We thank the referee for pointing out this omission. We agree with the referee that it is an important result for establishing the context of energy conditions in quantum theories and to also motivate the QEIs of study in our manuscript, and highlight their differences with the ANEC. We have added a discussion of the ANEC early in the introduction, starting in the second paragraph and continuing onward through the third paragraph. In the course of this discussion we have referenced more prior works on energy conditions and geometric theorems which we hope establishes a broader context for our work.

---

## Round 2 · List of Changes

A list of changes to the article (highlighted in blue) addressing referee 1’s comments are as follows:

1. Added footnote on page 3 clarifying what is meant by “smearing function.”
2. “diverges” to “divergences” in  line 196.
3. Added “local” to line 327.
4. Renamed state $|\psi\rangle$ to $|\Phi\rangle$ in equations (51) and (52).
5. Removed mention of the state $|h_\Delta^p\rangle$ from paragraph below equation (81).
6. Added sentence in line 344 on the assumption of no contact terms.

Changes to the most recent version of our manuscript are highlighted in blue and those addressing referee 2’s comments are listed as follows:

1. Expanded the introduction, starting at the middle of the second paragraph to establish the context of singularity theorems, ANEC, and QEIs.
2. Added paragraph under equation (6) on the role of state-dependent bounds in effective field theory.
3. Added sentence on ANEC in paragraph starting in line 131.
4. Added sentence in paragraph starting on line 188 about the consequences of the strong conjecture.
5. Added a new portion in the discussion section (starting line 864) establishing our results in the context of singularity theorems and wormhole spacetimes.

---

## Editorial Decision

published